# SurVAE Flows: Surjections to Bridge the Gap between VAEs and Flows

**Didrik Nielsen**[1], **Priyank Jaini**[2], **Emiel Hoogeboom**[2], **Ole Winther**[1], **Max Welling**[2]
Technical University of Denmark[1], UvA-Bosch Delta Lab, University of Amsterdam[2]
didni@dtu.dk, p.jaini@uva.nl, e.hoogeboom@uva.nl
olwi@dtu.dk, m.welling@uva.nl

## Abstract

Normalizing flows and variational autoencoders are powerful generative models that can represent complicated density functions. However, they both impose constraints on the models: Normalizing flows use bijective transformations to model densities whereas VAEs learn stochastic transformations that are non-invertible and thus typically do not provide tractable estimates of the marginal likelihood. In this paper, we introduce SurVAE Flows: A modular framework of composable transformations that encompasses VAEs and normalizing flows. SurVAE Flows bridge the gap between normalizing flows and VAEs with *surjective transformations*, wherein the transformations are deterministic in one direction – thereby allowing exact likelihood computation, and stochastic in the reverse direction – hence providing a lower bound on the corresponding likelihood. We show that several recently proposed methods, including dequantization and augmented normalizing flows, can be expressed as SurVAE Flows. Finally, we introduce common operations such as the *max value*, the *absolute value*, *sorting* and *stochastic permutation* as composable layers in SurVAE Flows.

## 1  Introduction

Normalizing flows (Tabak and Vanden-Eijnden, 2010; Tabak and Turner, 2013; Rezende and Mohamed, 2015) provide a powerful *modular* and *composable* framework for representing expressive probability densities via differentiable bijections (with a differentiable inverse). These composable bijective transformations accord significant advantages due to their ability to be implemented using a modular software framework with a general interface consisting of three important components: (i) a forward transform, (ii) an inverse transform, and (iii) a log-likelihood contribution through the Jacobian determinant. Thus, significant advances have been made in recent years to develop novel flow modules that are easily invertible, expressive and computationally cheap (Dinh et al., 2015, 2017; Kingma et al., 2016; Papamakarios et al., 2017; Huang et al., 2018; Jaini et al., 2019a,b; Kingma and Dhariwal, 2018; Hoogeboom et al., 2019b, 2020; Durkan et al., 2019; van den Berg et al., 2018).

However, the bijective nature of the transformations used for building normalizing flows limit their ability to alter dimensionality, model discrete data and distributions with discrete structure or disconnected components. Specialized solutions have been developed to address these limitations independently. Uria et al. (2013); Ho et al. (2019) use dequantization to model discrete distributions using continuous densities, while Tran et al. (2019); Hoogeboom et al. (2019a) propose a discrete analog of normalizing flows. Cornish et al. (2019) use an augmented space to model an infinite mixtures of normalizing flows to address the problem of disconnected components whereas Huang et al. (2020); Chen et al. (2020) use a similar idea of augmentation of the observation space to model expressive distributions. VAEs (Kingma and Welling, 2014; Rezende et al., 2014), on the other hand, have no such limitations, but only provide lower bound estimates of the tractable estimates for the

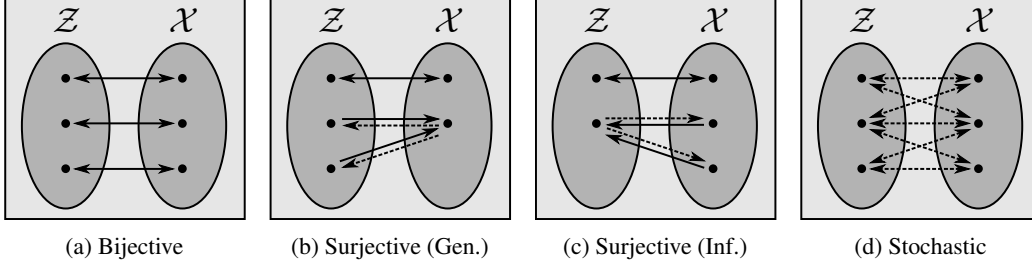

| (a) Bijective | (b) Surjective (Gen.) | (c) Surjective (Inf.) | (d) Stochastic |

Figure 1: Classes of SurVAE transformations $\mathcal{Z} \to \mathcal{X}$ and their inverses $\mathcal{X} \to \mathcal{Z}$. Solid lines indicate deterministic transformations, while dashed lines indicate stochastic transformations.

exact marginal density. These shortcomings motivate the question: *Is it possible to have composable and modular architectures that are expressive, model discrete and disconnected structure, and allow altering dimensions with exact likelihood evaluation?*

In this paper, we answer this affirmatively by introducing SurVAE Flows that use surjections to provide a unified, composable, and modular framework for probabilistic modeling. We introduce our unifying framework in §2 by identifying the components necessary to build composable architectures with modular software implementation for probabilistic modeling. We then introduce surjections for probabilistic modeling in §3 and show that these transformations lie at the interface between VAEs (stochastics maps) and normalizing flows (bijective maps). We unify these transformations (bijections, surjections, and stochastic transformations) in a composable and modular framework that we call SurVAE Flows. Subsequently, in §3.1, we propose novel SurVAE Flow layers like *max value* used for max pooling layers, *absolute value* for modelling symmetries in the data, and *sorting* and *stochastic permutations* that can be used for modelling exchangeable data and order statistics. Finally, in §3.2 we connect SurVAE Flows to several aforementioned specialised models by expressing them using SurVAE Flow layers which can now be implemented easily using our modular implementation. We demonstrate the efficacy of SurVAE Flows with experiments on synthetic datasets, point cloud data, and images. Code to implement SurVAE Flows and reproduce results is publicly available[1].

## 2 Preliminaries and Setup

In this section, we set up our main problem, provide key notations and definitions, and formulate a unifying framework for using different kinds of transformations to model distributions.

Let $x \in \mathcal{X}$ and $z \in \mathcal{Z}$ be two variables with distributions $p(x)$ and $p(z)$. We call a deterministic mapping $f : \mathcal{Z} \to \mathcal{X}$ *bijective* if it is both *surjective* and *injective*. A mapping is surjective if $\forall x \in \mathcal{X}$, $\exists z \in \mathcal{Z}$ such that $x = f(z)$. A mapping is injective if $\forall z_1, z_2 \in \mathcal{Z}, f(z_1) = f(z_2) \implies z_1 = z_2$. If the mapping is not deterministic, we refer to it as a stochastic mapping, and denote it as $z \sim p(x|z)$.

Normalizing flows (Tabak and Vanden-Eijnden, 2010; Tabak and Turner, 2013; Rezende and Mohamed, 2015) make use of bijective transformations $f$ to transform a simple base density $p(z)$ to a more expressive density $p(x)$, making using the change-of-variables formula $p(x) = p(z)|\det \nabla_x f^{-1}(x)|$. VAEs (Kingma and Welling, 2014; Rezende et al., 2014), on the other hand, define a probabilistic graphical model where each observed variable $x$ has an associated latent variable $z$ with the generative process as $z \sim p(z)$, $x \sim p(x|z)$, where $p(x|z)$ may be viewed as a stochastic transformation. VAEs use variational inference with an amortized variational distribution $q(z|x)$ to approximate the intractable posterior $p(z|x)$ which facilitates computation of a lower bound of $p(x)$ known as the evidence lower bound (ELBO) i.e., $\mathcal{L} := \mathbb{E}_{q(z|x)}[\log p(x|z)] - \mathbb{D}_{\mathsf{KL}}[q(z|x)\|p(z)]$.

In the following, we introduce a framework to connect flows and VAEs[2] by showing that bijective and stochastic transformations are *composable* and require three important components for use in probabilistic modeling: (i) a forward transformation, $f : \mathcal{Z} \to \mathcal{X}$ with an associated conditional probability $p(x|z)$, (ii) an inverse transformation, $f^{-1} : \mathcal{X} \to \mathcal{Z}$ with an associated distribution $q(z|x)$, and (iii) a *likelihood contribution* term used for log-likelihood computation.

Table 1: Composable building blocks of SurVAE Flows.

| Transformation | Forward $\boldsymbol{x} \leftarrow \boldsymbol{z}$ | Inverse $\boldsymbol{z} \leftarrow \boldsymbol{x}$ | Likelihood Contribution $\mathcal{V}(\boldsymbol{x}, \boldsymbol{z})$ | Bound Gap $\mathcal{E}(\boldsymbol{x}, \boldsymbol{z})$ |
|---|---|---|---|---|
| Bijective | $\boldsymbol{x} = f(\boldsymbol{z})$ | $\boldsymbol{z} = f^{-1}(\boldsymbol{x})$ | $\log|\det \nabla_{\boldsymbol{x}} \boldsymbol{z}|$ | $0$ |
| Stochastic | $\boldsymbol{x} \sim p(\boldsymbol{x}|\boldsymbol{z})$ | $\boldsymbol{z} \sim q(\boldsymbol{z}|\boldsymbol{x})$ | $\log \frac{p(\boldsymbol{x}|\boldsymbol{z})}{q(\boldsymbol{z}|\boldsymbol{x})}$ | $\log \frac{q(\boldsymbol{z}|\boldsymbol{x})}{p(\boldsymbol{z}|\boldsymbol{x})}$ |
| Surjective (Gen.) | $\boldsymbol{x} = f(\boldsymbol{z})$ | $\boldsymbol{z} \sim q(\boldsymbol{z}|\boldsymbol{x})$ | $\log \frac{p(\boldsymbol{x}|\boldsymbol{z})}{q(\boldsymbol{z}|\boldsymbol{x})}$ as $\begin{smallmatrix}p(\boldsymbol{x}|\boldsymbol{z}) \to \\ \delta(\boldsymbol{x} - f(\boldsymbol{z}))\end{smallmatrix}$ | $\log \frac{q(\boldsymbol{z}|\boldsymbol{x})}{p(\boldsymbol{z}|\boldsymbol{x})}$ |
| Surjective (Inf.) | $\boldsymbol{x} \sim p(\boldsymbol{x}|\boldsymbol{z})$ | $\boldsymbol{z} = f^{-1}(\boldsymbol{x})$ | $\log \frac{p(\boldsymbol{x}|\boldsymbol{z})}{q(\boldsymbol{z}|\boldsymbol{x})}$ as $\begin{smallmatrix}q(\boldsymbol{z}|\boldsymbol{x}) \to \\ \delta(\boldsymbol{z} - f^{-1}(\boldsymbol{x}))\end{smallmatrix}$ | $0$ |

**Forward Transformation:** For a stochastic transformation, the forward transformation is the conditional distribution $p(\boldsymbol{x}|\boldsymbol{z})$. For a bijective transformation, on the other hand, the forward transformation is deterministic and therefore, $p(\boldsymbol{x}|\boldsymbol{z}) = \delta(\boldsymbol{x} - f(\boldsymbol{z}))$ or simply $\boldsymbol{x} = f(\boldsymbol{z})$.

**Inverse Transformation:** For a bijective transformation, the inverse is also deterministic and given by $\boldsymbol{z} = f^{-1}(\boldsymbol{x})$. For a stochastic transformation, the inverse is also stochastic and is defined by Bayes theorem $p(\boldsymbol{z}|\boldsymbol{x}) = p(\boldsymbol{x}|\boldsymbol{z})p(\boldsymbol{z})/p(\boldsymbol{x})$. Computing $p(\boldsymbol{z}|\boldsymbol{x})$ is typically intractable and we thus resort to a variational approximation $q(\boldsymbol{z}|\boldsymbol{x})$.

**Likelihood Contribution:** For bijections, the density $p(\boldsymbol{x})$ can be computed from $p(\boldsymbol{z})$ and the mapping $f$ using the change-of-variables formula as:

$$\log p(\boldsymbol{x}) = \log p(\boldsymbol{z}) + \log|\det \boldsymbol{J}|, \qquad \boldsymbol{z} = f^{-1}(\boldsymbol{x}) \tag{1}$$

where $|\det \boldsymbol{J}| = |\det \nabla_{\boldsymbol{x}} f^{-1}(\boldsymbol{x})|$ is the absolute value of the determinant of the Jacobian matrix of $f^{-1}$ which defines the likelihood contribution term for a bijective transformation $f$. For stochastic transformations, we can rewrite the marginal density $p(\boldsymbol{x})$ as:

$$\log p(\boldsymbol{x}) = \underbrace{\mathbb{E}_{q(\boldsymbol{z}|\boldsymbol{x})}[\log p(\boldsymbol{x}|\boldsymbol{z})] - \mathbb{D}_{\mathsf{KL}}[q(\boldsymbol{z}|\boldsymbol{x})\|p(\boldsymbol{z})]}_{\text{ELBO}} + \underbrace{\mathbb{D}_{\mathsf{KL}}[q(\boldsymbol{z}|\boldsymbol{x})\|p(\boldsymbol{z}|\boldsymbol{x})]}_{\text{Gap in Lower Bound}} \tag{2}$$

The ELBO $\mathcal{L}$ in Eq. 2 can then be evaluated using a single Monte Carlo sample: $\mathcal{L} \approx \log p(\boldsymbol{z}) + \log \frac{p(\boldsymbol{x}|\boldsymbol{z})}{q(\boldsymbol{z}|\boldsymbol{x})}$, $\boldsymbol{z} \sim q(\boldsymbol{z}|\boldsymbol{x})$. Therefore, the likelihood contribution term for a stochastic transformation is defined as $\log \frac{p(\boldsymbol{x}|\boldsymbol{z})}{q(\boldsymbol{z}|\boldsymbol{x})}$. Furthermore, we show in App. A that Eq. 2 allows us to recover the change-of-variables formula given in Eq. 1 by using Dirac delta functions, thereby drawing a precise connection between VAEs and normalizing flows. Crucially, Eq. 2 helps us to reveal a unified modular framework to model a density $p(\boldsymbol{x})$ under any transformation by restating it as:

$$\log p(\boldsymbol{x}) \simeq \log p(\boldsymbol{z}) + \mathcal{V}(\boldsymbol{x}, \boldsymbol{z}) + \mathcal{E}(\boldsymbol{x}, \boldsymbol{z}), \quad \boldsymbol{z} \sim q(\boldsymbol{z}|\boldsymbol{x}) \tag{3}$$

where $\mathcal{V}(\boldsymbol{x}, \boldsymbol{z})$ and $\mathcal{E}(\boldsymbol{x}, \boldsymbol{z})$ are the *likelihood contribution* and *bound looseness* terms, respectively. The likelihood contribution is $\mathcal{V}(\boldsymbol{x}, \boldsymbol{z}) = \log|\det \boldsymbol{J}|$ for bijections and $\log \frac{p(\boldsymbol{x}|\boldsymbol{z})}{q(\boldsymbol{z}|\boldsymbol{x})}$ for stochastic transformations. For bijections, likelihood evaluation is deterministic and *exact* with $\mathcal{E}(\boldsymbol{x}, \boldsymbol{z}) = 0$, while for stochastic maps it is stochastic and unbiased with a bound looseness of $\mathcal{E}(\boldsymbol{x}, \boldsymbol{z}) = \log \frac{q(\boldsymbol{z}|\boldsymbol{x})}{p(\boldsymbol{z}|\boldsymbol{x})}$. This is summarized in Table 1. The first term in Eq. 3, $\log p(\boldsymbol{z})$, reveals the compositional nature of the transformations, since it can be modeled by further transformations. While the compositional structure has been used widely for bijective transformations, Eq. 3 demonstrates its viability for stochastic maps as well. We demonstrate this unified compositional structure in Alg. 1.

---

**Algorithm 1:** $\log - \text{likelihood}(\boldsymbol{x})$

**Data:** $\boldsymbol{x}$, $p(\boldsymbol{z})$ & $\{f_t\}_{t=1}^T$
**Result:** $\mathcal{L}(\boldsymbol{x})$
**for** $t$ *in* $\text{range}(T)$, **do**
    **if** $f_t$ *is bijective* **then**
        $\boldsymbol{z} = f_t^{-1}(\boldsymbol{x})$ ;
        $\mathcal{V}_t = \log\left|\det \frac{\partial \boldsymbol{z}}{\partial \boldsymbol{x}}\right|$ ;
    **else if** $f_t$ *is stochastic* **then**
        $\boldsymbol{z} \sim q_t(\boldsymbol{z}|\boldsymbol{x})$ ;
        $\mathcal{V}_t = \log \frac{p_t(\boldsymbol{x}|\boldsymbol{z})}{q_t(\boldsymbol{z}|\boldsymbol{x})}$ ;
    $\boldsymbol{x} = \boldsymbol{z}$ ;
**end**
**return** $\log p(\boldsymbol{z}) + \sum_{t=1}^T \mathcal{V}_t$

---

# 3 SurVAE Flows

As explained in Section 2, bijective and stochastic transformations provide a modular framework for constructing expressive generative models. However, they both impose constraints on the model: bijective transformations are deterministic and allow exact likelihood computation, but they are required to preserve dimensionality. On the other hand, stochastic transformations are capable of altering the dimensionality of the random variables but only provide a stochastic lower bound estimate of the likelihood. *Is it possible to have composable transformations that can alter dimensionality and allow exact likelihood evaluation?* In this section, we answer this question affirmatively by introducing surjective transformations as SurVAE Flows that bridge the gap between bijective and stochastic transformations.

In the following, we will define composable deterministic transformations that are surjective and non-injective. For brevity, we will refer to them as *surjections* or *surjective transformations*. Note that for surjections, multiple inputs can map to a single output, resulting in a *loss of information* since the input is not guaranteed to be recovered through inversion. Similar to bijective and stochastic transformations, the three important components of composable surjective transformations are:

**Forward Transformation:** Like bijections, surjective transformations have a deterministic forward transformation $p(\boldsymbol{x}|\boldsymbol{z}) = \delta\big(\boldsymbol{x} - f(\boldsymbol{z})\big)$ or $\boldsymbol{x} = f(\boldsymbol{z})$.

**Inverse Transformation:** In contrast with bijections, surjections $f : \mathcal{Z} \to \mathcal{X}$ are not invertible since multiple inputs can map to the same output. However, they have *right inverses*, i.e. functions $g : \mathcal{X} \to \mathcal{Z}$ such that $f \circ g(\boldsymbol{x}) = \boldsymbol{x}$, but not necessarily $g \circ f(\boldsymbol{z}) = \boldsymbol{z}$. We will use a stochastic right inverse $q(\boldsymbol{z}|\boldsymbol{x})$ which can be thought of as passing $\boldsymbol{x}$ through a random right inverse $g$. Importantly, $q(\boldsymbol{z}|\boldsymbol{x})$ only has support over the preimage of $\boldsymbol{x}$, i.e. the set of $\boldsymbol{z}$ that map to $\boldsymbol{x}$, $\mathcal{B}(\boldsymbol{x}) = \{\boldsymbol{z}|\boldsymbol{x} = f(\boldsymbol{z})\}$.

So far, we have described what we will term *generative surjections*, i.e. transformations that are surjective in the generative direction $\mathcal{Z} \to \mathcal{X}$. We will refer to a transformation which is surjective in the inference direction $\mathcal{X} \to \mathcal{Z}$ as an *inference surjection*. These are illustrated in Fig.1. Generative surjections have stochastic inverse transformations $q(\boldsymbol{z}|\boldsymbol{x})$, while inference surjections have stochastic forward $p(\boldsymbol{x}|\boldsymbol{z})$ transformations.

**Likelihood Contribution:** For continuous surjections, the likelihood contribution term is:

$$\mathbb{E}_{q(\boldsymbol{z}|\boldsymbol{x})}\left[\log \frac{p(\boldsymbol{x}|\boldsymbol{z})}{q(\boldsymbol{z}|\boldsymbol{x})}\right], \quad \text{as} \quad \begin{cases} p(\boldsymbol{x}|\boldsymbol{z}) \to \delta\big(\boldsymbol{x} - f(\boldsymbol{z})\big), & \text{for gen. surjections.} \\ q(\boldsymbol{z}|\boldsymbol{x}) \to \delta\big(\boldsymbol{z} - f^{-1}(\boldsymbol{x})\big), & \text{for inf. surjections.} \end{cases}$$

While generative surjections generally give rise to stochastic estimates of the likelihood contribution and introduce lower bound likelihood estimates, inference surjections allow *exact likelihood computation* (see App. B). Before proceeding further, we give a few examples to better understand the construction of a surjective transformation for probabilistic modeling.

**Example 1 (Tensor slicing)** *Let $f$ be a tensor slicing surjection that takes input $\boldsymbol{z} = (\boldsymbol{z}_1, \boldsymbol{z}_2) \in \mathbb{R}^{d_z}$ and returns a subset of the elements, i.e. $\boldsymbol{x} = f(\boldsymbol{z}) = \boldsymbol{z}_1$. To develop this operation as a SurVAE layer, we first specify the stochastic forward and inverse transformations as:*

$$p(\boldsymbol{x}|\boldsymbol{z}) = \mathcal{N}(\boldsymbol{x}|\boldsymbol{z}_1, \sigma^2 \boldsymbol{I}), \quad \text{and} \quad q(\boldsymbol{z}|\boldsymbol{x}) = \mathcal{N}(\boldsymbol{z}_1|\boldsymbol{x}, \sigma^2 \boldsymbol{I})q(\boldsymbol{z}_2|\boldsymbol{x})$$

*We next compute the likelihood contribution term in the limit that $p(\boldsymbol{x}|\boldsymbol{z}) \to \delta\big(\boldsymbol{x} - f(\boldsymbol{z})\big)$. Here, this corresponds to $\sigma \to 0$. Thus,*

$$\mathcal{V}(\boldsymbol{x}, \boldsymbol{z}) = \lim_{\sigma^2 \to 0} \mathbb{E}_{q(\boldsymbol{z}|\boldsymbol{x})}\left[\log \frac{p(\boldsymbol{x}|\boldsymbol{z})}{q(\boldsymbol{z}|\boldsymbol{x})}\right] = \mathbb{E}_{q(\boldsymbol{z}_2|\boldsymbol{x})}\left[-\log q(\boldsymbol{z}_2|\boldsymbol{x})\right],$$

*which corresponds to the entropy of $q(\boldsymbol{z}_2|\boldsymbol{x})$ that is used to infer the sliced elements $\boldsymbol{z}_2$. We illustrate the slicing surjection for both the generative and inference directions in Fig. 2.*

**Example 2 (Rounding)** *Let $f$ be a rounding surjection that takes an input $\boldsymbol{z} \in \mathbb{R}^{d_z}$ and returns the rounded $\boldsymbol{x} := \lfloor \boldsymbol{z} \rfloor$. The forward transformation is a discrete surjection $P(\boldsymbol{x}|\boldsymbol{z}) = \mathbb{I}(\boldsymbol{z} \in \mathcal{B}(\boldsymbol{x}))$, for $\mathcal{B}(\boldsymbol{x}) = \{\boldsymbol{x} + \boldsymbol{u}|\boldsymbol{u} \in [0, 1)^d\}$. The inverse transformation $q(\boldsymbol{z}|\boldsymbol{x})$ is stochastic with support in $\mathcal{B}(\boldsymbol{x})$. Inserting this in the likelihood contribution term and simplifying, we find*

$$\mathcal{V}(\boldsymbol{x}, \boldsymbol{z}) = \mathbb{E}_{q(\boldsymbol{z}|\boldsymbol{x})}\left[-\log q(\boldsymbol{z}|\boldsymbol{x})\right].$$

*This generative rounding surjection gives rise to dequantization (Uria et al., 2013; Ho et al., 2019) which is a method commonly used to train continuous flows on discrete data such as images.*

Table 2: Summary of selected inference surjection layers. See App. C for more SurVAE layers.

| Surjection | Forward | Inverse | $\mathcal{V}(\boldsymbol{x}, \boldsymbol{z})$ |
|---|---|---|---|
| Abs | $s \sim \text{Bern}(\pi(z))$ <br> $x = s \cdot z, \ s \in \{-1, 1\}$ | $s = \text{sign}\, x$ <br> $z = \lvert x \rvert$ | $\log p(s\lvert z)$ |
| Max | $k \sim \text{Cat}(\boldsymbol{\pi}(z))$ <br> $x_k = z, \boldsymbol{x}_{-k} \sim p(\boldsymbol{x}_{-k}\lvert z, k)$ | $k = \arg\max \boldsymbol{x}$ <br> $z = \max \boldsymbol{x}$ | $\log p(k\lvert z) + \log p(\boldsymbol{x}_{-k}\lvert z, k)$ |
| Sort | $\mathcal{I} \sim \text{Cat}(\boldsymbol{\pi}(\boldsymbol{z}))$ <br> $\boldsymbol{x} = \boldsymbol{z}_{\mathcal{I}}$ | $\mathcal{I} = \text{argsort}\, \boldsymbol{x}$ <br> $\boldsymbol{z} = \text{sort}\, \boldsymbol{x}$ | $\log p(\mathcal{I}\lvert \boldsymbol{z})$ |

The preceding discussion shows that surjective transformations can be composed to construct expressive transformations for density modelling. We call a single surjective transformation a SurVAE layer and a composition of bijective, surjective, and/or stochastic transformations a SurVAE Flow. The unified framework of SurVAE Flows allows us to construct generative models learned using the likelihood (or its lower bound) of the data, utilizing Eq. 3, and Table 1.

## 3.1 Novel SurVAE Layers

We developed the tensor slicing and rounding surjections in Examples 1 and 2. In this section, we introduce additional novel SurVAE layers including the *absolute value*, the *maximum value* and *sorting* as surjective layers and *stochastic permutation* as a stochastic layer. We provide a summary of these in Table 2. Due to space constraints, we defer the derivations and details on each of these surjections to Appendix D-G along with detailed tables on generative and inference surjections in Table 6 and 7.

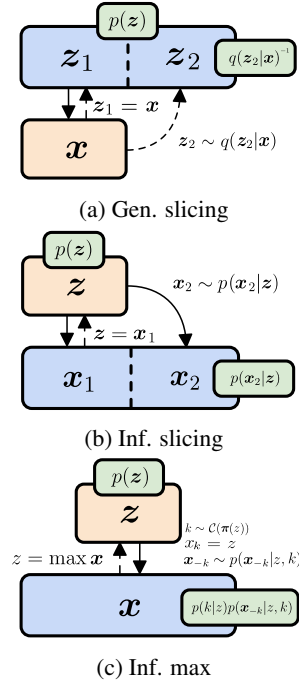

(a) Gen. slicing

(b) Inf. slicing

(c) Inf. max

Figure 2: Surjections.

**Abs Surjection** (App. D). The abs surjection returns the the magnitude of its input, $z = \lvert x \rvert$. As a SurVAE layer, we can represent the inference surjection with the forward and inverse transformations as:

$$p(x\lvert z) = \sum_{s \in \{-1,1\}} p(\boldsymbol{x}\lvert z, s) p(s\lvert z) = \sum_{s \in \{-1,1\}} \delta(x - sz) p(s\lvert z),$$

$$q(z\lvert x) = \sum_{s \in \{-1,1\}} q(z\lvert \boldsymbol{x}, s) p(s\lvert x) = \sum_{s \in \{-1,1\}} \delta(z - sx) \delta_{s, \text{sign}(x)}$$

where $q(z\lvert x)$ is deterministic corresponding to $z = \lvert x \rvert$. The forward transformation $p(x\lvert z)$ involves the following steps: (i) sample the sign $s$, conditioned on $z$, and (ii) apply the sign to $z$ to obtain $x = sz$. Abs surjections are useful for modelling data with symmetries which we demonstrate in our experiments.

**Max Surjection** (App. E, Fig. 2). The max operator returns the largest element of an input vector, $z = \max \boldsymbol{x}$. We can represent this transformation as

$$p(\boldsymbol{x}\lvert z) = \sum_{k=1}^{K} p(\boldsymbol{x}\lvert z, k) p(k\lvert z) = \sum_{k=1}^{K} \delta(x_k - z) p(\boldsymbol{x}_{-k}\lvert z, k) p(k\lvert z),$$

$$q(z\lvert \boldsymbol{x}) = \sum_{k=1}^{K} q(z\lvert \boldsymbol{x}, k) q(k\lvert x) = \sum_{k=1}^{K} \delta(z - x_k) \delta_{k, \arg\max(\boldsymbol{x})},$$

where $q(z\lvert x)$ is deterministic and corresponds to $z = \max \boldsymbol{x}$. While the inverse is deterministic The stochastic forward proceeds by (i) sampling an index $k$ and setting $x_k = z$, and (ii) imputing the remaining values $\boldsymbol{x}_{-k}$ of $\boldsymbol{x}$ such that they are all smaller than $x_k$. Max surjections are useful in implementing the *max pooling* layer commonly used in convolutional architectures for downsampling. In our experiments, we demonstrate the use of max surjections for probabilistic modelling of images.

**Sort Surjection** (App. F). Sorting, $\boldsymbol{z} = \text{sort}(\boldsymbol{x})$ returns a vector in sorted order. It is a surjective (and non-injective) transformation since the original order of the vector is lost in the operation even

though the dimensions remain the same. Sort surjections are useful in modelling naturally sorted data, learning order statistics, and learning an exchangeable model using flows.

**Stochastic Permutation** (App. G). A stochastic permutation transforms the input vector by shuffling the elements randomly. The inverse pass for a permutation is the same as the forward pass with the likelihood contribution term equal to zero, $\mathcal{V} = 0$. Stochastic permutations helps to enforce permutation invariance i.e. any flow can be made permutation invariant by adding a final permutation SurVAE layer. In our experiments, we compare sorting surjections and stochastic permutations to enforce permutation invariance for modelling exchangeable data.

**Stochastic Inverse Parameterization**. For surjections, the stochastic inverses have to be defined so that they form a distribution over the possible right-inverses. Different right-inverse distributions do not have to align over subsets of the space. Consequently, for more sophisticated choices of right-inverses, the log-likelihood may be discontinuous across boundaries of these subsets. Since these points have measure zero, this does not influence the validity of the log-likelihood. However, it may impede optimization using gradient-based methods. In our experiments, we did not encounter any specific issues, but for a more thorough discussion see (Dinh et al., 2019).

## 3.2 Connection to Previous Work

The results above provide a unified framework based on SurVAE Flows for estimating probability densities. We now connect this general approach to several recent works on generative modelling.

The differentiable and bijective nature of transformations used in normalizing flows limit their ability to alter dimensionality, model discrete data, and distributions with disconnected components. Specialized solutions have been proposed in recent years to address these individually. We now show that these works can be expressed using SurVAE Flow layers, as summarized in Table 3.

### 3.2.1 Using Stochastic Transformations

As discussed in Sec. 2, VAEs (Kingma and Welling, 2014; Rezende et al., 2014) may be formulated as composable stochastic transformations. Probabilistic PCA Tipping and Bishop (1999) can be considered a simple special case of VAEs wherein the forward transformation is linear-Gaussian, i.e. $p(\boldsymbol{x}|\boldsymbol{z}) = \mathcal{N}(\boldsymbol{x}|\boldsymbol{W}\boldsymbol{z}, \sigma^2 \boldsymbol{I})$. Due to the linear-Gaussian formulation, the posterior $p(\boldsymbol{z}|\boldsymbol{x})$ is tractable and we can thus perform exact stochastic inversion for this model. Diffusion models (Sohl-Dickstein et al., 2015; Ho et al., 2020) are another class of models closely related to VAEs. For diffusion models, the inverse $q(\boldsymbol{z}|\boldsymbol{x})$ implements a diffusion step, while the forward transformation $p(\boldsymbol{x}|\boldsymbol{z})$ learns to reverse the diffusion process. Wu et al. (2020) propose an extended flow framework consisting of bijective and stochastic transformations using MCMC transition kernels. Their method utilizes the same computation as in the general formulation in Algorithm 1, but does not consider surjective maps or an explicit connection to VAEs. Their work shows that MCMC kernels may also be implemented as stochastic transformations in SurVAE Flows.

Table 3: SurVAE Flows as a unifying framework.

| Model | SurVAE Flow architecture |
| --- | --- |
| Probabilistic PCA (Tipping and Bishop, 1999) <br> VAE (Kingma and Welling, 2014; Rezende et al., 2014) <br> Diffusion Models (Sohl-Dickstein et al., 2015; Ho et al., 2020) | $\mathcal{Z} \xrightarrow{\text{stochastic}} \mathcal{X}$ |
| Dequantization (Uria et al., 2013; Ho et al., 2019) | $\mathcal{Z} \xrightarrow{\text{round}} \mathcal{X}$ |
| ANFs, VFlow (Huang et al., 2020; Chen et al., 2020) | $\mathcal{X} \xrightarrow{\text{augment}} \mathcal{X} \times \mathcal{E} \xrightarrow{\text{bijection}} \mathcal{Z}$ |
| Multi-scale Architectures (Dinh et al., 2017) | $\mathcal{X} \xrightarrow{\text{bijection}} \mathcal{Y} \times \mathcal{E} \xrightarrow{\text{slice}} \mathcal{Y} \xrightarrow{\text{bijection}} \mathcal{Z}$ |
| CIFs, Discretely Indexed Flows, DeepGMMs <br> (Cornish et al., 2019; Duan, 2019; Oord and Dambre, 2015) | $\mathcal{X} \xrightarrow{\text{augment}} \mathcal{X} \times \mathcal{E} \xrightarrow{\text{bijection}} \mathcal{Z} \times \mathcal{E} \xrightarrow{\text{slice}} \mathcal{Z}$ |
| RAD Flows (Dinh et al., 2019) | $\mathcal{X} \xrightarrow{\text{partition}} \mathcal{X}_{\mathcal{E}} \times \mathcal{E} \xrightarrow{\text{bijection}} \mathcal{Z} \times \mathcal{E} \xrightarrow{\text{slice}} \mathcal{Z}$ |

### 3.2.2 Using Surjective Transformations

Dequantization (Uria et al., 2013; Ho et al., 2019) is used for training continuous flow models on ordinal discrete data such as images and audio. Dequantization fits into the SurVAE Flow framework as a composable generative rounding surjection (cf. Example 2) and thus simplifies implementation. When the inverse $q(\boldsymbol{z}|\boldsymbol{x})$ is a standard uniform distributon, *uniform dequantization* is obtained, while a more flexible flow-based distribution $q(\boldsymbol{z}|\boldsymbol{x})$ yields *variational dequantization* (Ho et al., 2019).

VFlow (Chen et al., 2020) and ANFs (Huang et al., 2020) aim to build expressive generative models by augmenting the data space and jointly learning a normalizing flow for the augmented data space as well as the distribution of augmented dimensions. This strategy was also adopted by Dupont et al. (2019) for continuous-time flows. VFlow and ANFs can be obtained as SurVAE Flows by composing a bijection with a generative tensor slicing surjection (cf. Example 1 and Figure 3a). The reverse transformation, i.e. inference slicing, results in the *multi-scale architecture* of Dinh et al. (2017).

CIFs (Cornish et al., 2019) use an indexed family of bijective transformations $g(\cdot; \varepsilon) : \mathcal{Z} \to \mathcal{X}$ where $\mathcal{Z} = \mathcal{X} \subseteq \mathbb{R}^d$, and $\varepsilon \in \mathcal{E} \subseteq \mathbb{R}^{d_\varepsilon}$ with the generative process as: $\boldsymbol{z} \sim p(\boldsymbol{z})$, $\epsilon \sim p(\epsilon|\boldsymbol{z})$ and $\boldsymbol{x} = g(\boldsymbol{z}; \varepsilon)$ and requires specifying $p(\boldsymbol{z})$ and $p(\varepsilon|\boldsymbol{z})$. CIFs are akin to modeling densities using an infinite mixture of normalizing flows since $g$ is a surjection from an augmented space $\mathcal{Z} \times \mathcal{E}$ to the data space $\mathcal{X}$. Consequently, CIFs can be expressed as a SurVAE flow using a augment surjection composed with a bijection and tensor slicing (cf. Figure 3b). Similarly, Duan (2019) used a *finite* mixture of normalizing flows to model densities by using a discrete index set $\mathcal{E} = \{1, 2, 3, \cdots, K\}$ with bijections $g(\cdot; \varepsilon)$. Deep Gaussian mixture models (Oord and Dambre, 2015) form special case wherein the bijections $g(\cdot; \varepsilon)$ are linear transformations.

RAD flows (Dinh et al., 2019) are also "similar" to CIFs but it partitions the data space into finite disjoint subsets $\{\mathcal{X}_i\}_{i=1}^K \subseteq \mathcal{X}$ and defines bijections $g_i : \mathcal{Z} \to \mathcal{X}_i, \forall\, i \in \{1, 2, ..., K\}$ with the generative process as $\boldsymbol{z} \sim p(\boldsymbol{z}), i \sim p(i|\boldsymbol{z})$ and $\boldsymbol{x} = g_i(\boldsymbol{z})$. Interestingly, RAD can be seen to implement a class of inference surjections that rely on partitioning of the data space. The partitioning is learned during training, thus allowing learning of expressive inference surjections. However, careful parameterization is required for stable gradient-based training. We note that the abs and sort inference surjections introduced earlier may be expressed using a static (non-learned) partitioning of the data space $\mathcal{X}$ and thus have close ties to RAD. However, RAD does not express generative surjections or more general inference surjections that do not rely on partitioning, such as dimensional changes.

Finally, we note that apart from providing a general method for modelling densities, SurVAE Flows provide a modular framework for easy implementation of the methods described here. We discuss these important software perspectives using code snippets in App. H.

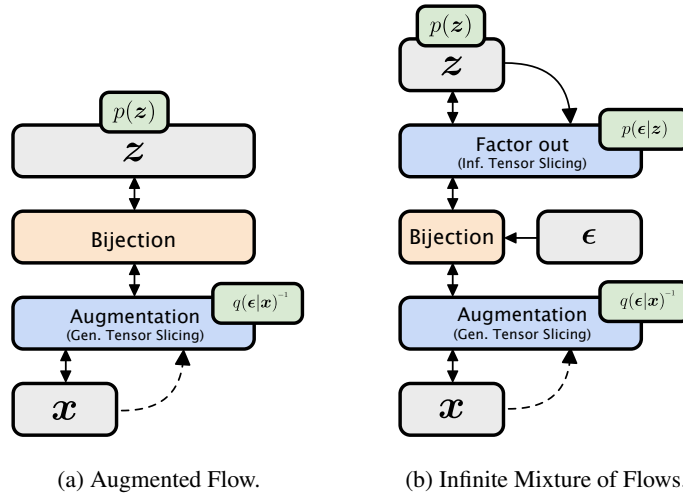

(a) Augmented Flow.     (b) Infinite Mixture of Flows.

Figure 3: Flow architectures making use of tensor slicing.

# 4 Experiments

We investigate the ability of SurVAE flows to model data that is difficult to model with normalizing flows. We show that the absolute value surjection is useful in modelling data where certain symmetries are known to exist. Next, we demonstrate that SurVAE flows allow straightforward modelling of exchangeable data by simply composing *any* flow together with either a sorting surjection or a stochastic permutation layer. Furthermore, we investigate the use of max pooling – which is commonly used for downsampling in convolutional neural networks – as a surjective downsampling layer in SurVAE flows for image data.

**Synthetic Data.** We first consider modelling data where certain symmetries are known to exist. We make use of 3 symmetric and 1 anti-symmetric synthetic 2D datasets. The absolute value inference surjection can be seen to fold the input space across the origin and can thus be useful in modelling such data. The baseline uses 4 coupling bijections, while our *AbsFlow* adds an extra `abs` surjection. For the anti-symmetric data, AbsFlow uses only a single `abs` surjection with a classifier (*i.e.* for $P(s|z)$) which learns the unfolding. For further details, see App. I.1. The results are shown in Fig. 4.

**Point Cloud Data.** We now consider modelling exchangeable data. We use the `SpatialMNIST` dataset (Edwards and Storkey, 2017), where each MNIST digit is represented as a 2D point cloud of 50 points. A point cloud is a set, *i.e.* it is permutation invariant. Using SurVAE flows, we can enforce permutation invariance on *any* flow using either 1) a sorting surjection – forcing a canonical order on the inputs, or 2) a stochastic permutation – forcing a random order on the inputs.

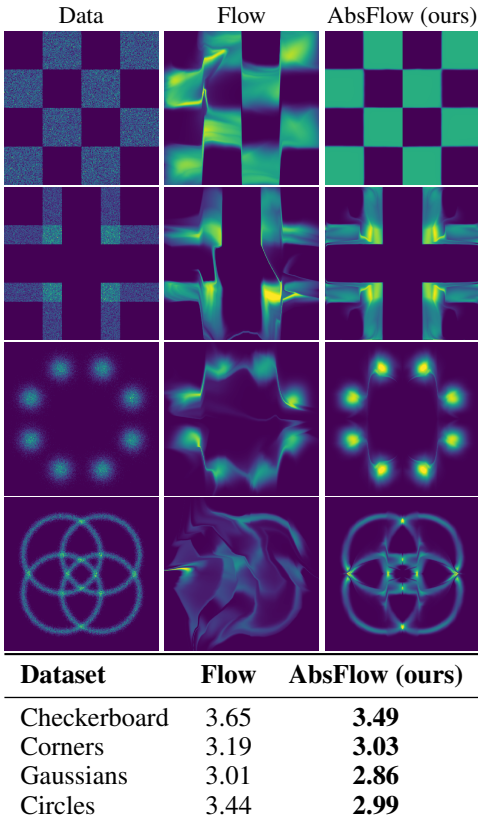

| Data | Flow | AbsFlow (ours) |
| --- | --- | --- |

| Dataset | Flow | AbsFlow (ours) |
| --- | --- | --- |
| Checkerboard | 3.65 | **3.49** |
| Corners | 3.19 | **3.03** |
| Gaussians | 3.01 | **2.86** |
| Circles | 3.44 | **2.99** |

Figure 4: Comparison of flows with and without absolute value surjections modelling anti-symmetric (top row) and symmetric (3 bottom rows) 2-dimensional distributions.

We compare 2 SurVAE flows, *SortFlow* and *PermuteFlow*, both using 64 layers of coupling flows parameterized by Transformer networks (Vaswani et al., 2017). Transformers are – when *not* using positional encoding – permutation equivariant. PermuteFlow uses stochastic permutation in-between the coupling layers. SortFlow, on the other hand, uses and initial sorting surjection, which introduces an ordering, and fixed permutations after. The Transformers thus make use of learned positional encodings for SortFlow, but not for PermuteFlow. See App. I.2 for further details, and Fig. 5 for

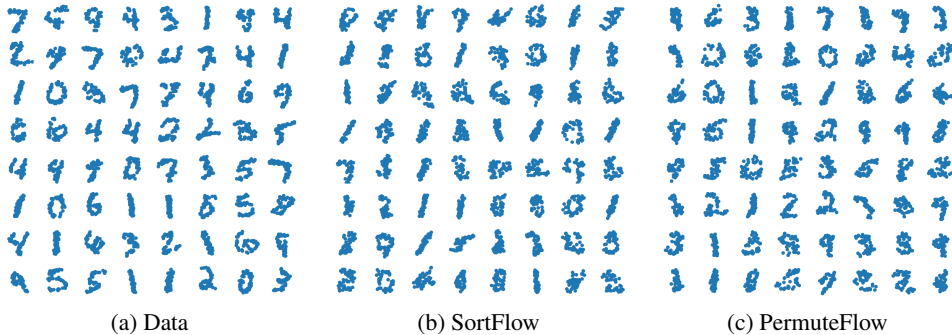

|          (a) Data          |         (b) SortFlow         |        (c) PermuteFlow         |

Figure 5: Point cloud samples from permutation-invariant SurVAE flows trained on `SpatialMNIST`.

Table 4: Unconditional image modeling results in bits/dim.

| Model | CIFAR-10 | ImageNet32 | ImageNet64 |
|---|---|---|---|
| RealNVP (Dinh et al., 2017) | 3.49 | 4.28 | - |
| Glow (Kingma and Dhariwal, 2018) | 3.35 | 4.09 | 3.81 |
| Flow++ (Ho et al., 2019) | 3.08 | 3.86 | 3.69 |
| Baseline (Ours) | **3.08** | **4.00** | **3.70** |
| MaxPoolFlow (Ours) | 3.09 | 4.01 | 3.74 |

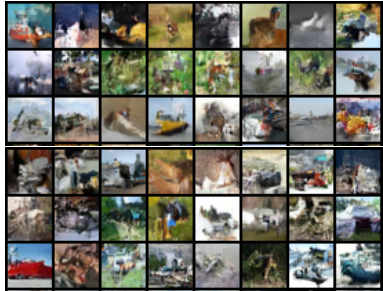

Figure 7: Samples from CIFAR-10 models. Top: MaxPoolFlow, Bottom: Baseline.

| Model | Inception ↑ | FID ↓ |
|---|---|---|
| DCGAN* | 6.4 | 37.1 |
| WGAN-GP* | 6.5 | 36.4 |
| PixelCNN* | 4.60 | 65.93 |
| PixelIQN* | 5.29 | 49.46 |
| Baseline (Ours) | 5.08 | 49.56 |
| MaxPoolFlow (Ours) | **5.18** | **49.03** |

Table 5: Inception score and FID for CIFAR-10. *Results taken from Ostrovski et al. (2018).

model samples. Interestingly, PermuteFlow outperforms SortFlow, with -5.30 vs. -5.53 PPLL (per-point log-likelihood), even though it only allows computation of lower bound likelihood estimates. For comparison, BRUNO (Korshunova et al., 2018) and FlowScan (Bender et al., 2020) obtain -5.68 and -5.26 PPLL, but make use of autoregressive components. Neural Statistican (Edwards and Storkey, 2017) utilizes hierarchical latent variables without autoregressive parts and obtains -5.37 PPL. PermuteFlow thus obtains *state-of-the-art* performance among non-autoregressive models.

**Image Data.** Max pooling layers are commonly used for downsampling in convolutional neural networks. We investigate their use as surjective downsampling transformations in flow models for image data here.

We train a flow using 2 scales with 12 steps/scale for `CIFAR-10` and `ImageNet 32×32` and 3 scales with 8 steps/scale for `ImageNet 64×64`. Each step consists of an affine coupling bijection and a $1×1$ convolution (Kingma and Dhariwal, 2018). We implement a max pooling surjection for downscaling and compare it to a baseline model with tensor slicing which corresponds to a *multi-scale* architecture (Dinh et al., 2017). We report results for the log-likelihood in Table 4 and the inception and FID scores in Table 5 with bolding indicating best among the baseline and MaxPoolFlow. The results show that compared to slicing surjections, the max pooling surjections yield marginally worse log-likelihoods, but better visual sample quality as measured by the Inception score and FID. We also provide the generated samples from our models in Fig. 7 and App. J. Due to space constrains, we refer the reader to App. I.3 for more details on the experiment.

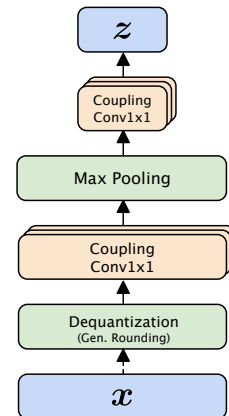

Figure 6: Flow architecture with max pooling. Surjections in green.

## 5 Conclusion

We introduced SurVAE flows, a modular framework for constructing likelihood-based models using composable bijective, surjective and stochastic transformations. We showed how this encompasses normalizing flows, which rely on bijections, as well as VAEs, which rely on stochastic transformations. We further showed that several recently proposed methods such as dequantization and augmented normalizing flows may be obtained as SurVAE flows using surjective transformations. One interesting direction for further research is development of novel non-bijective transformations that might be beneficial as composable layers in SurVAE flows.

## Acknowledgements

We thank Laurent Dinh and Giorgio Giannone for helpful feedback.

## Funding Disclosure

This research was supported by the NVIDIA Corporation with the donation of TITAN X GPUs.

## Broader Impact

This work constitutes foundational research on generative models/unsupervised learning by providing a unified view on several lines of work and further by introducing new modules that expand the generative modelling toolkit. This work further suggests how to build software libraries to that allows more rapid implementation of a wider range of deep unsupervised models. Unsupervised learning has the potential to greatly reduce the need for labeled data and thus improve models in applications such as medical imaging where a lack of data can be a limitation. However, it may also potentially be used to improve deep fakes with potentially malicious applications.

## Footnotes

[1] The code is available at `https://github.com/didriknielsen/survae_flows`

[2] We note that Wu et al. (2020) also considered stochastic maps in flows using MCMC transition kernels.

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
