[Supplementary Material]

# A  A Connection Between VAEs and Flows

Variational Autoencoders (VAEs) can be seen as a composable stochastic transformations. From this viewpoint, the log-likelihood resulting from a single transformation can be written as

$$\log p(\boldsymbol{x}) = \mathbb{E}_{q(\boldsymbol{z}|\boldsymbol{x})}\left[\log p(\boldsymbol{z})\right] + \underbrace{\mathbb{E}_{q(\boldsymbol{z}|\boldsymbol{x})}\left[\log \frac{p(\boldsymbol{x}|\boldsymbol{z})}{q(\boldsymbol{z}|\boldsymbol{x})}\right]}_{\text{Lik. contrib. } \mathcal{V}(\boldsymbol{x},\boldsymbol{z})} + \underbrace{\mathbb{E}_{q(\boldsymbol{z}|\boldsymbol{x})}\left[\log \frac{q(\boldsymbol{z}|\boldsymbol{x})}{p(\boldsymbol{z}|\boldsymbol{x})}\right]}_{\text{Bound looseness } \mathcal{E}(\boldsymbol{x},\boldsymbol{z})}, \tag{4}$$

which consists of *1)* the log-likelihood of $\boldsymbol{z} \sim q(\boldsymbol{z}|\boldsymbol{x})$ under the remaining layers $p(\boldsymbol{z})$, *2)* the likelihood contribution term $\mathcal{V}(\boldsymbol{x}, \boldsymbol{z})$ and *3)* the looseness of the bound $\mathcal{E}(\boldsymbol{x}, \boldsymbol{z})$.

Normalizing flows, on the other hand, make use of deterministic transformations. Specifically, using a diffeomorphism $f : \mathcal{Z} \to \mathcal{X}$, the log-likelihood can be computed as

$$\log p(\boldsymbol{x}) = \log p(\boldsymbol{z}) + \underbrace{\log\left|\det\frac{\partial \boldsymbol{z}}{\partial \boldsymbol{x}}\right|}_{\text{Lik. contrib. } \mathcal{V}(\boldsymbol{x},\boldsymbol{z})}, \quad \boldsymbol{z} = f^{-1}(\boldsymbol{x}), \tag{5}$$

which consists *1)* the log-likelihood of $\boldsymbol{z} = f^{-1}(\boldsymbol{x})$ under $p(\boldsymbol{z})$ (possibly another flow) and *2)* the likelihood contribution term, which here corresponds to the log Jacobian determinant. Notice that for normalizing flows, the likelihood is exact and hence the *bound looseness* term $\mathcal{E}(\boldsymbol{x}, \boldsymbol{z}) = 0$.

In the remainder of this section we show that the change-of-variables formula (Eq. 5) can be obtained from the ELBO (Eq. 4).

**Proof.** We can use a composition of a function $g$ with a Dirac $\delta$-function:

$$\int \delta(g(\boldsymbol{z}))f(g(\boldsymbol{z}))\left|\det\frac{\partial g(\boldsymbol{z})}{\partial \boldsymbol{z}}\right|d\boldsymbol{z} = \int \delta(\boldsymbol{u})f(\boldsymbol{u})d\boldsymbol{u} \tag{6}$$

to conclude that

$$\delta(g(\boldsymbol{z})) = \left|\det\frac{\partial g(\boldsymbol{z})}{\partial \boldsymbol{z}}\right|^{-1}_{\boldsymbol{z}=\boldsymbol{z}_0}\delta(\boldsymbol{z} - \boldsymbol{z}_0) \tag{7}$$

with $\boldsymbol{z}_0$ being the root of $g(\boldsymbol{z})$. This results assumes that $g$ is smooth (derivative exists), $f$ has compact support, the root is unique and the Jacobian is non-singular.

Let $f : \mathcal{Z} \to \mathcal{X}$ be a diffeomorphism and define a pair of deterministic conditionals

$$p(\boldsymbol{x}|\boldsymbol{z}) = \delta(\boldsymbol{x} - f(\boldsymbol{z})) \tag{8}$$

$$p(\boldsymbol{z}|\boldsymbol{x}) = \delta(\boldsymbol{z} - f^{-1}(\boldsymbol{x})). \tag{9}$$

Applying the above result to $p(\boldsymbol{x}|\boldsymbol{z})$, we set $g(\boldsymbol{z}) = \boldsymbol{x} - f(\boldsymbol{z})$ and find $\boldsymbol{z}_0 = f^{-1}(\boldsymbol{x})$ and

$$p(\boldsymbol{x}|\boldsymbol{z}) = \delta(\boldsymbol{z} - f^{-1}(\boldsymbol{x}))|\det \boldsymbol{J}| = p(\boldsymbol{z}|\boldsymbol{x})|\det \boldsymbol{J}|, \tag{10}$$

where

$$\boldsymbol{J}^{-1} = \left.\frac{\partial f(\boldsymbol{z})}{\partial \boldsymbol{z}}\right|_{\boldsymbol{z}=f^{-1}(\boldsymbol{x})}.$$

Let further $q(\boldsymbol{z}|\boldsymbol{x}) = p(\boldsymbol{z}|\boldsymbol{x}) = \delta(\boldsymbol{z} - f^{-1}(\boldsymbol{x}))$. The resulting ELBO gives rise to the change-of-variables formula,

$$\log p(\boldsymbol{x}) = \mathbb{E}_{q(\boldsymbol{z}|\boldsymbol{x})}\left[\log p(\boldsymbol{z}) + \log\frac{p(\boldsymbol{x}|\boldsymbol{z})}{q(\boldsymbol{z}|\boldsymbol{x})} + \log\frac{q(\boldsymbol{z}|\boldsymbol{x})}{p(\boldsymbol{z}|\boldsymbol{x})}\right] \tag{11}$$

$$= \log p(\boldsymbol{z}) + \log|\det \boldsymbol{J}|, \quad \text{for } \boldsymbol{z} = f^{-1}(\boldsymbol{x}), \tag{12}$$

where the likelihood contribution $\mathcal{V}(\boldsymbol{x}, \boldsymbol{z}) = \log\frac{p(\boldsymbol{x}|\boldsymbol{z})}{q(\boldsymbol{z}|\boldsymbol{x})} = \log|\det \boldsymbol{J}|$, while the bound looseness term $\mathcal{E}(\boldsymbol{x}, \boldsymbol{z}) = \log\frac{q(\boldsymbol{z}|\boldsymbol{x})}{p(\boldsymbol{z}|\boldsymbol{x})} = 0$, trivially.

## B    The Bound Looseness for Inference Surjections

For inference surjections $f : \mathcal{X} \to \mathcal{Z}$, the bound looseness term $\mathcal{E}(\boldsymbol{x}, \boldsymbol{z}) = 0$, given that the *stochastic right inverse condition* is satified. The stochastic right inverse condition requires that $p(\boldsymbol{x}|\boldsymbol{z})$ defines a distribution over the possible right inverses of the surjection $f$.

A right inverse function $g : \mathcal{Z} \to \mathcal{X}$ to a function $f : \mathcal{X} \to \mathcal{Z}$ satisfies $f \circ g = \mathrm{id}_{\mathcal{Z}}$, but not necessarily $g \circ f = \mathrm{id}_{\mathcal{X}}$. Here $\mathrm{id}_{\mathcal{S}}$ denotes an identity map defined on the space $\mathcal{S}$.

We satisfy the stochastic right inverse condition by requiring that $p(\boldsymbol{x}|\boldsymbol{z})$ only has support over the *fiber* of $\boldsymbol{z}$, i.e. the set of elements $\mathcal{B}(\boldsymbol{z})$ in the domain $\mathcal{X}$ that are mapped to $\boldsymbol{z}$, $\mathcal{B}(\boldsymbol{z}) := \{\boldsymbol{x}|\boldsymbol{z} = f(\boldsymbol{x})\}$. A simple check for stochastic right invertibility is thus: For any $\boldsymbol{z}$, computing $\boldsymbol{z} = f(\boldsymbol{x})$, for $\boldsymbol{x} \sim p(\boldsymbol{x}|\boldsymbol{z})$ should return the original $\boldsymbol{z}$.

Given that the distribution $p(\boldsymbol{z})$ has full support over $\mathcal{Z}$ and the stochastic right inverse condition is satisfied, we have that, for any observed $\boldsymbol{x}$, only one $\boldsymbol{z}$ could have given rise to the observation $\boldsymbol{x}$. Consequently, the posterior distribution $p(\boldsymbol{z}|\boldsymbol{x}) = \delta(\boldsymbol{z} - f(\boldsymbol{x}))$ is deterministic. By defining $q(\boldsymbol{z}|\boldsymbol{x}) = p(\boldsymbol{z}|\boldsymbol{x})$, the bound looseness is thus $\mathcal{E}(\boldsymbol{x}, \boldsymbol{z}) = 0$.

## C    List of SurVAE Layers

See Table 6 and Table 7 for lists of generative and inference surjection layers, respectively.

Table 6: Summary of some generative surjection layers.

| Surjection | Forward | Inverse | $\mathcal{V}(\boldsymbol{x}, \boldsymbol{z})$ |
|---|---|---|---|
| Rounding | $x = \lfloor z \rfloor$ | $z \sim q(z|x)$ where $z \in [x, x+1)$ | $-\log q(z|x)$ |
| Slicing | $\boldsymbol{x} = \boldsymbol{z}_1$ | $\boldsymbol{z}_1 = \boldsymbol{x}, \boldsymbol{z}_2 \sim q(\boldsymbol{z}_2|\boldsymbol{x})$ | $-\log q(\boldsymbol{z}_2|\boldsymbol{x})$ |
| Abs | $s = \mathrm{sign}\, z$ $x = |z|$ | $s \sim \mathrm{Bern}(\pi(x))$ $z = s \cdot x,\ \ s \in \{1, -1\}$ | $-\log q(s|x)$ |
| Max | $k = \arg\max \boldsymbol{z}$ $x = \max \boldsymbol{z}$ | $k \sim \mathrm{Cat}(\boldsymbol{\pi}(x))$ $z_k = x, \boldsymbol{z}_{-k} \sim q(\boldsymbol{z}_{-k}|x, k)$ | $-\log q(k|x) - \log q(\boldsymbol{z}_{-k}|x, k)$ |
| Sort | $\mathcal{I} = \mathrm{argsort}\, \boldsymbol{z}$ $\boldsymbol{x} = \mathrm{sort}\, \boldsymbol{z}$ | $\mathcal{I} \sim \mathrm{Cat}(\boldsymbol{\pi}(\boldsymbol{x}))$ $\boldsymbol{z} = \boldsymbol{x}_{\mathcal{I}}$ | $-\log q(\mathcal{I}|\boldsymbol{x})$ |
| ReLU | $x = \max(z, 0)$ | if $x = 0 : z \sim q(z)$, else $: z = x$ | $\mathbb{I}(x = 0)[-\log q(z)]$ |

Table 7: Summary of some inference surjection layers.

| Surjection | Forward | Inverse | $\mathcal{V}(\boldsymbol{x}, \boldsymbol{z})$ |
|---|---|---|---|
| Rounding | $x \sim p(x|z)$ where $x \in [z, z+1)$ | $z = \lfloor x \rfloor$ | $\log p(z|x)$ |
| Slicing | $\boldsymbol{x}_1 = \boldsymbol{z}, \boldsymbol{x}_2 \sim p(\boldsymbol{x}_2|\boldsymbol{z})$ | $\boldsymbol{z} = \boldsymbol{x}_1$ | $\log p(\boldsymbol{x}_2|\boldsymbol{z})$ |
| Abs | $s \sim \mathrm{Bern}(\pi(z))$ $x = s \cdot z,\ \ s \in \{-1, 1\}$ | $s = \mathrm{sign}\, x$ $z = |x|$ | $\log p(s|z)$ |
| Max | $k \sim \mathrm{Cat}(\boldsymbol{\pi}(z))$ $x_k = z, \boldsymbol{x}_{-k} \sim p(\boldsymbol{x}_{-k}|z, k)$ | $k = \arg\max \boldsymbol{x}$ $z = \max \boldsymbol{x}$ | $\log p(k|z) + \log p(\boldsymbol{x}_{-k}|z, k)$ |
| Sort | $\mathcal{I} \sim \mathrm{Cat}(\boldsymbol{\pi}(\boldsymbol{z}))$ $\boldsymbol{x} = \boldsymbol{z}_{\mathcal{I}}$ | $\mathcal{I} = \mathrm{argsort}\, \boldsymbol{x}$ $\boldsymbol{z} = \mathrm{sort}\, \boldsymbol{x}$ | $\log p(\mathcal{I}|\boldsymbol{z})$ |
| ReLU | if $z = 0 : x \sim p(x)$, else $: x = z$ | $z = \max(x, 0)$ | $\mathbb{I}(z = 0) \log p(x)$ |

# D The Absolute Value Surjection

We here develop the absolute value surjections, both in the generative direction $x = |z|$ and in the inference direction $z = |x|$. We will make use of Dirac delta functions to develop the likelihood contributions, but we could equivalently develop them using Gaussian distributions where $\sigma \to 0$.

## D.1 Generative Direction

**Forward and Inverse.** We define the forward and inverse transformations as

$$p(x|z) = \sum_{s \in \{-1,1\}} p(x|z,s)p(s|z) = \sum_{s \in \{-1,1\}} \delta(x - sz)\delta_{s,\text{sign}(z)}, \tag{13}$$

$$q(z|x) = \sum_{s \in \{-1,1\}} q(z|x,s)q(s|x) = \sum_{s \in \{-1,1\}} \delta(z - sx)q(s|x), \tag{14}$$

where the forward transformation $p(x|z)$ is fully deterministic and corresponds to $x = |z|$. The inference direction involves two steps, 1) sample the sign $s$ of $z$ conditioned of $x$, and 2) deterministically map $x$ to $z = sx$. Note that $q(s|x)$ may either be trained as a classifier or fixed to e.g. $q(s|x) = 1/2$. The last choice especially makes sense when $p(z)$ is symmetric.

**Likelihood Contribution.** We may develop the likelihood contribution by computing

$$\mathcal{V} = \mathbb{E}_{q(z|x,s)q(s|x)} \left[ \log \frac{p(x|z,s)p(s|z)}{q(z|x,s)q(s|x)} \right] \tag{15}$$

$$= \mathbb{E}_{\delta(z-sx)q(s|x)} \left[ \log \frac{\delta(x - sz)\delta_{s,\text{sign}(z)}}{\delta(z - sx)q(s|x)} \right] \tag{16}$$

$$\approx -\log q(s|x), \quad \text{where } z = sx, \ s \sim q(s|x). \tag{17}$$

Here, $\delta(x - sz)$ and $\delta(z - sx)$ cancel since $\delta(x - sz) = \delta(z - x/s)|1/s| = \delta(z - sx)$.

## D.2 Inference Direction

**Forward and Inverse.** We define the forward and inverse transformations as

$$p(x|z) = \sum_{s \in \{-1,1\}} p(x|z,s)p(s|z) = \sum_{s \in \{-1,1\}} \delta(x - sz)p(s|z), \tag{18}$$

$$q(z|x) = \sum_{s \in \{-1,1\}} q(z|x,s)q(s|x) = \sum_{s \in \{-1,1\}} \delta(z - sx)\delta_{s,\text{sign}(x)}, \tag{19}$$

where the inverse transformation $q(z|x)$ is fully deterministic and corresponds to $z = |x|$. The generative direction involves two steps, 1) sample the sign of $x$ conditioned of $z$, and 2) deterministically map $z$ to $x = sz$. Note that $p(s|z)$ may either be trained as a classifier or fixed to e.g. $p(s|z) = 1/2$. The last choice gives rise to an absolute value surjection which may be used to enforce exact symmetry across the origin.

**Likelihood Contribution.** We may develop the likelihood contribution by computing

$$\mathcal{V} = \mathbb{E}_{q(z|x,s)q(s|x)} \left[ \log \frac{p(x|z,s)p(s|z)}{q(z|x,s)q(s|x)} \right] \tag{20}$$

$$= \mathbb{E}_{\delta(z-sx)\delta_{s,\text{sign}(x)}} \left[ \log \frac{\delta(x - sz)p(s|z)}{\delta(z - sx)\delta_{s,\text{sign}(x)}} \right] \tag{21}$$

$$= \log p(s|z), \quad \text{where } z = sx = |x|, \ s = \text{sign}(x). \tag{22}$$

Here, $\delta(x - sz)$ and $\delta(z - sx)$ cancel since $\delta(x - sz) = \delta(z - x/s)|1/s| = \delta(z - sx)$.

# E   The Maximum Value Surjection

We here develop the maximum value surjections, both in the generative direction $x = \max \boldsymbol{z}$ and in the inference direction $z = \max \boldsymbol{x}$. We will make use of Dirac delta functions to develop the likelihood contributions, but we could equivalently develop them using Gaussian distributions where $\sigma \to 0$.

## E.1   Generative Direction

**Forward and Inverse.** We define the forward and inverse transformations as

$$p(x|\boldsymbol{z}) = \sum_{k=1}^{K} p(x|\boldsymbol{z}, k)p(k|\boldsymbol{z}) = \sum_{k=1}^{K} \delta(x - z_k)\delta_{k, \arg\max(\boldsymbol{z})}, \tag{23}$$

$$q(\boldsymbol{z}|x) = \sum_{k=1}^{K} q(\boldsymbol{z}|x, k)q(k|x) = \sum_{k=1}^{K} \delta(z_k - x)q(\boldsymbol{z}_{-k}|x, k)q(k|x), \tag{24}$$

where $k$ refers to the indices of $\boldsymbol{z}$, $K$ is the number of elements in $\boldsymbol{z}$ and $\boldsymbol{z}_{-k}$ is $\boldsymbol{z}$ excluding element $k$. The forward transformation $p(x|\boldsymbol{z})$ is fully deterministic and corresponds to $x = \max \boldsymbol{z}$. The inference direction involves three steps, 1) sample the index $k$ for the argmax of $\boldsymbol{z}$ conditioned of $x$, 2) deterministically map $x$ to $z_k = x$, and 3) infer the remaining elements $\boldsymbol{z}_{-k}$ of $\boldsymbol{z}$. Note that $q(k|x)$ may either be trained as a classifier or fixed to e.g. $q(k|x) = 1/K$.

For $q$ to define a right-inverse of $p$, we require that $q(\boldsymbol{z}_{-k}|x, k)$ only has support in $(-\infty, x)^{K-1}$ such that $z_k$ will be the maximum value.

**Likelihood Contribution.** We may develop the likelihood contribution by computing

$$\mathcal{V} = \mathbb{E}_{q(\boldsymbol{z}|x,k)q(k|x)} \left[ \log \frac{p(x|\boldsymbol{z}, k)p(k|\boldsymbol{z})}{q(\boldsymbol{z}|x, k)q(k|x)} \right] \tag{25}$$

$$= \mathbb{E}_{\delta(z_k - x)q(\boldsymbol{z}_{-k}|x,k)q(k|x)} \left[ \log \frac{\delta(x - z_k)\delta_{k, \arg\max(\boldsymbol{z})}}{\delta(z_k - x)q(\boldsymbol{z}_{-k}|x, k)q(k|x)} \right] \tag{26}$$

$$\approx -\log q(k|x) - \log q(\boldsymbol{z}_{-k}|x, k), \quad \text{where } z_k = x, \ \boldsymbol{z}_{-k} \sim q(\boldsymbol{z}_{-k}|x, k), \ k \sim q(k|x). \tag{27}$$

## E.2   Inference Direction

**Forward and Inverse.** We define the forward and inverse transformations as

$$p(\boldsymbol{x}|z) = \sum_{k=1}^{K} p(\boldsymbol{x}|z, k)p(k|z) = \sum_{k=1}^{K} \delta(x_k - z)p(\boldsymbol{x}_{-k}|z, k)p(k|z), \tag{28}$$

$$q(z|\boldsymbol{x}) = \sum_{k=1}^{K} q(z|\boldsymbol{x}, k)q(k|x) = \sum_{k=1}^{K} \delta(z - x_k)\delta_{k, \arg\max(\boldsymbol{x})}, \tag{29}$$

where $k$ refers to the indices of $\boldsymbol{x}$, $K$ is the number of elements in $\boldsymbol{x}$ and $\boldsymbol{x}_{-k}$ is $\boldsymbol{x}$ excluding element $k$. The inverse transformation $q(z|\boldsymbol{x})$ is fully deterministic and corresponds to $z = \max \boldsymbol{x}$. The inference direction involves three steps, 1) sample the index $k$ for the argmax of $\boldsymbol{x}$ conditioned of $z$, 2) deterministically map $z$ to $x_k = z$, and 3) infer the remaining elements $\boldsymbol{x}_{-k}$ of $\boldsymbol{x}$. Note that $p(k|z)$ may either be trained as a classifier or fixed to e.g. $p(k|z) = 1/K$.

For $p$ to define a right-inverse of $q$, we require that $p(\boldsymbol{x}_{-k}|z, k)$ only has support in $(-\infty, z)^{K-1}$ such that $x_k$ will be the maximum value.

**Likelihood Contribution.** We may develop the likelihood contribution by computing

$$\mathcal{V} = \mathbb{E}_{q(z|\boldsymbol{x},k)q(k|\boldsymbol{x})} \left[ \log \frac{p(\boldsymbol{x}|z, k)p(k|z)}{q(z|\boldsymbol{x}, k)q(k|\boldsymbol{x})} \right] \tag{30}$$

$$= \mathbb{E}_{\delta(z - x_k)\delta_{k, \arg\max(\boldsymbol{x})}} \left[ \log \frac{\delta(x_k - z)p(\boldsymbol{x}_{-k}|z, k)p(k|z)}{\delta(z - x_k)\delta_{k, \arg\max(\boldsymbol{x})}} \right] \tag{31}$$

$$= \log p(k|z) + \log p(\boldsymbol{x}_{-k}|z, k), \quad \text{where } z = x_k = \max \boldsymbol{x}, \ k = \arg\max \boldsymbol{x}. \tag{32}$$

# F The Sort Surjection

We here develop the sorting surjections, both in the generative direction $x = \text{sort}\,z$ and in the inference direction $z = \text{sort}\,x$. We will make use of Dirac delta functions to develop the likelihood contributions, but we could equivalently develop them using Gaussian distributions where $\sigma \to 0$.

## F.1 Generative Direction

**Forward and Inverse.** We define the forward and inverse transformations as

$$p(\boldsymbol{x}|\boldsymbol{z}) = \sum_{\mathcal{I}} p(\boldsymbol{x}|\boldsymbol{z}, \mathcal{I}) p(\mathcal{I}|\boldsymbol{z}) = \sum_{\mathcal{I}} \delta(\boldsymbol{x} - \boldsymbol{z}_{\mathcal{I}}) \delta_{\mathcal{I}, \text{argsort}(\boldsymbol{z})}, \tag{33}$$

$$q(\boldsymbol{z}|\boldsymbol{x}) = \sum_{\mathcal{I}} q(\boldsymbol{z}|\boldsymbol{x}, \mathcal{I}) q(\mathcal{I}|\boldsymbol{x}) = \sum_{\mathcal{I}} \delta(\boldsymbol{z} - \boldsymbol{x}_{\mathcal{I}^{-1}}) q(\mathcal{I}|\boldsymbol{x}), \tag{34}$$

where $\mathcal{I}$ refers to a set of permutation indices, $\mathcal{I}^{-1}$ refers to the inverse permutation indices and $\boldsymbol{z}_{\mathcal{I}}$ refers to the elements of $\boldsymbol{z}$ permuted according to the indices $\mathcal{I}$. Note that there are $D!$ possible permutations.

The forward transformation $p(\boldsymbol{x}|\boldsymbol{z})$ is fully deterministic and corresponds to $x = \text{sort}\,z$. The inference direction involves two steps, 1) sample permutation indices $\mathcal{I}$ conditioned of $\boldsymbol{x}$, and 2) deterministically permute $\boldsymbol{x}$ according to the inverse permutation $\mathcal{I}^{-1}$ to obtain $\boldsymbol{z} = \boldsymbol{x}_{\mathcal{I}^{-1}}$. Note that $q(\mathcal{I}|\boldsymbol{x})$ may either be trained as a classifier or fixed to e.g. $q(\mathcal{I}|\boldsymbol{x}) = 1/D!$.

**Likelihood Contribution.** We may develop the likelihood contribution by computing

$$\mathcal{V} = \mathbb{E}_{q(\boldsymbol{z}|\boldsymbol{x}, \mathcal{I}) q(\mathcal{I}|\boldsymbol{x})} \left[ \log \frac{p(\boldsymbol{x}|\boldsymbol{z}, \mathcal{I}) p(\mathcal{I}|\boldsymbol{z})}{q(\boldsymbol{z}|\boldsymbol{x}, \mathcal{I}) q(\mathcal{I}|\boldsymbol{x})} \right] \tag{35}$$

$$= \mathbb{E}_{\delta(\boldsymbol{z} - \boldsymbol{x}_{\mathcal{I}^{-1}}) q(\mathcal{I}|\boldsymbol{x})} \left[ \log \frac{\delta(\boldsymbol{x} - \boldsymbol{z}_{\mathcal{I}}) \delta_{\mathcal{I}, \text{argsort}(\boldsymbol{z})}}{\delta(\boldsymbol{z} - \boldsymbol{x}_{\mathcal{I}^{-1}}) q(\mathcal{I}|\boldsymbol{x})} \right] \tag{36}$$

$$\approx -\log q(\mathcal{I}|\boldsymbol{x}), \quad \text{where } \mathcal{I} \sim q(\mathcal{I}|\boldsymbol{x}). \tag{37}$$

## F.2 Inference Direction

**Forward and Inverse.** We define the forward and inverse transformations as

$$p(\boldsymbol{x}|\boldsymbol{z}) = \sum_{\mathcal{I}} p(\boldsymbol{x}|\boldsymbol{z}, \mathcal{I}) p(\mathcal{I}|\boldsymbol{z}) = \sum_{\mathcal{I}} \delta(\boldsymbol{x} - \boldsymbol{z}_{\mathcal{I}^{-1}}) p(\mathcal{I}|\boldsymbol{z}), \tag{38}$$

$$q(\boldsymbol{z}|\boldsymbol{x}) = \sum_{\mathcal{I}} q(\boldsymbol{z}|\boldsymbol{x}, \mathcal{I}) q(\mathcal{I}|\boldsymbol{x}) = \sum_{\mathcal{I}} \delta(\boldsymbol{z} - \boldsymbol{x}_{\mathcal{I}}) \delta_{\mathcal{I}, \text{argsort}(\boldsymbol{x})}, \tag{39}$$

where $\mathcal{I}$ refers to a set of permutation indices, $\mathcal{I}^{-1}$ refers to the inverse permutation indices and $\boldsymbol{x}_{\mathcal{I}}$ refers to the elements of $\boldsymbol{x}$ permuted according to the indices $\mathcal{I}$. Note that there are $D!$ possible permutations.

The inverse transformation $q(\boldsymbol{z}|\boldsymbol{x})$ is fully deterministic and corresponds to $z = \text{sort}\,x$. The generative direction involves two steps, 1) sample permutation indices $\mathcal{I}$ conditioned of $\boldsymbol{z}$, and 2) deterministically permute $\boldsymbol{z}$ according to the inverse permutation $\mathcal{I}^{-1}$ to obtain $\boldsymbol{x} = \boldsymbol{z}_{\mathcal{I}^{-1}}$. Note that $p(\mathcal{I}|\boldsymbol{z})$ may either be trained as a classifier or fixed to e.g. $p(\mathcal{I}|\boldsymbol{z}) = 1/D!$.

**Likelihood Contribution.** We may develop the likelihood contribution by computing

$$\mathcal{V} = \mathbb{E}_{q(\boldsymbol{z}|\boldsymbol{x}, \mathcal{I}) q(\mathcal{I}|\boldsymbol{x})} \left[ \log \frac{p(\boldsymbol{x}|\boldsymbol{z}, \mathcal{I}) p(\mathcal{I}|\boldsymbol{z})}{q(\boldsymbol{z}|\boldsymbol{x}, \mathcal{I}) q(\mathcal{I}|\boldsymbol{x})} \right] \tag{40}$$

$$= \mathbb{E}_{\delta(\boldsymbol{z} - \boldsymbol{x}_{\mathcal{I}}) \delta_{\mathcal{I}, \text{argsort}(\boldsymbol{x})}} \left[ \log \frac{\delta(\boldsymbol{x} - \boldsymbol{z}_{\mathcal{I}^{-1}}) p(\mathcal{I}|\boldsymbol{z})}{\delta(\boldsymbol{z} - \boldsymbol{x}_{\mathcal{I}}) \delta_{\mathcal{I}, \text{argsort}(\boldsymbol{x})}} \right] \tag{41}$$

$$= \log p(\mathcal{I}|\boldsymbol{z}), \quad \text{where } \boldsymbol{z} = \boldsymbol{x}_{\mathcal{I}} = \text{sort}\,x, \ \mathcal{I} = \text{argsort}\,x. \tag{42}$$

# G The Stochastic Permutation

We here develop the stochastic permutation layer which randomly permutes its input. The inverse pass mirrors the forward pass. Note that stochastic permutation is *not* a surjection, but rather a stochastic transform. We will make use of Dirac delta functions to develop the likelihood contributions, but we could equivalently develop them using Gaussian distributions where $\sigma \to 0$.

**Forward and Inverse.** We define the forward and inverse transformations as

$$p(\boldsymbol{x}|\boldsymbol{z}) = \sum_{\mathcal{I}} p(\boldsymbol{x}|\boldsymbol{z}, \mathcal{I}) p(\mathcal{I}) = \sum_{\mathcal{I}} \delta(\boldsymbol{x} - \boldsymbol{z}_{\mathcal{I}}) \operatorname{Unif}(\mathcal{I}), \tag{43}$$

$$q(\boldsymbol{z}|\boldsymbol{x}) = \sum_{\mathcal{I}} q(\boldsymbol{z}|\boldsymbol{x}, \mathcal{I}) q(\mathcal{I}) = \sum_{\mathcal{I}} \delta(\boldsymbol{z} - \boldsymbol{x}_{\mathcal{I}^{-1}}) \operatorname{Unif}(\mathcal{I}), \tag{44}$$

where $\mathcal{I}$ refers to a set of permutation indices, $\mathcal{I}^{-1}$ refers to the inverse permutation indices and $\boldsymbol{z}_{\mathcal{I}}$ refers to the elements of $\boldsymbol{z}$ permuted according to the indices $\mathcal{I}$. Note that there are $D!$ possible permutations.

The transformation is stochastic and involves the same two steps in both directions: 1) Sample permutation indices $\mathcal{I}$ uniformly at random, and 2) deterministically permute the input according to the samples indices $\mathcal{I}$.

**Likelihood Contribution.** We may develop the likelihood contribution by computing

$$\mathcal{V} = \mathbb{E}_{q(\boldsymbol{z}|\boldsymbol{x}, \mathcal{I}) q(\mathcal{I})} \left[ \log \frac{p(\boldsymbol{x}|\boldsymbol{z}, \mathcal{I}) p(\mathcal{I})}{q(\boldsymbol{z}|\boldsymbol{x}, \mathcal{I}) q(\mathcal{I})} \right] \tag{45}$$

$$= \mathbb{E}_{\delta(\boldsymbol{z} - \boldsymbol{x}_{\mathcal{I}^{-1}}) \operatorname{Unif}(\mathcal{I})} \left[ \log \frac{\delta(\boldsymbol{x} - \boldsymbol{z}_{\mathcal{I}}) \operatorname{Unif}(\mathcal{I})}{\delta(\boldsymbol{z} - \boldsymbol{x}_{\mathcal{I}^{-1}}) \operatorname{Unif}(\mathcal{I})} \right] \tag{46}$$

$$= 0. \tag{47}$$

This layer thus takes the simple form: Both during the forward and inverse passes, shuffle the input uniformly at random. The resulting likelihood contribution is zero.

# H The Software Perspective

Normalizing flows provide a powerful modular framework where flexible densities may be specified using a composition of bijective transformations. Each bijection may be implemented as a module contained 3 important components: 1) A forward transformation $\boldsymbol{x} = f(\boldsymbol{z})$, 2) an inverse transformation $\boldsymbol{z} = f^{-1}(\boldsymbol{x})$, and 3) a Jacobian determinant $\log|\det \boldsymbol{J}|$. Several software libraries for normalizing flows have been built using this modular design principle (Dillon et al., 2017; Bingham et al., 2018).

SurVAE flows suggest that such software frameworks may be directly extended since the modules follow the exact same design principles – each module has 3 important components:

1. A forward transformation $\mathcal{Z} \rightarrow \mathcal{X}$.
2. An inverse transformation $\mathcal{X} \rightarrow \mathcal{Z}$.
3. A likelihood contribution $\mathcal{V}(\boldsymbol{x}, \boldsymbol{z})$.

SurVAE flows allow compositions of not only bijective transformations, but also surjective and stochastic transformations. This allows us to obtain methods such as dequantization (Uria et al., 2014; Theis et al., 2016; Ho et al., 2019), variational data augmentation (Huang et al., 2020; Chen et al., 2020), multi-scale architectures (Dinh et al., 2017) as composable surjective transformations and VAEs (Kingma and Welling, 2014; Rezende et al., 2014) as composable stochastic transformations.

In our code[3], we provide a library of SurVAE flows that may serve as a prototype for a more extensive library. In the next subsections, we show some selected code snippets from our library. The code is based on PyTorch (Paszke et al., 2019), but can easily be ported to other frameworks. Note that in the implementation, the `forward` method implements the inverse transformation $\mathcal{X} \rightarrow \mathcal{Z}$ and the likelihood contribution $\mathcal{V}(\boldsymbol{x}, \boldsymbol{z})$, since this is what is needed during the forward pass of backpropagation used for training.

In Sec. H.1 we show an implementation of a VAE as a stochastic transformation, while in Sec. H.2 and Sec. H.3 we show implementations of dequantization and variational data augmentation as surjective transformations. Finally, in Sec. H.4, we show an example of how to construct an augmented normalizing flow through composition of SurVAE layers.

## H.1 VAE

We implement VAEs as a composable stochastic transformation.

```
class VAE(StochasticTransform):
    '''A variational autoencoder layer.'''

    def __init__(self, decoder, encoder):
        super(VAE, self).__init__()
        self.decoder = decoder
        self.encoder = encoder

    def forward(self, x):
        z, log_qz = self.encoder.sample_with_log_prob(context=x)
        log_px = self.decoder.log_prob(x, context=z)
        ldj = log_px - log_qz
        return z, ldj

    def inverse(self, z):
        x = self.decoder.sample(context=z)
        return x
```

## H.2  Dequantization

We implement `UniformDequantization`, which may be used to convert between discrete and continuous variables, as a generative rounding surjection.

```python
class UniformDequantization(Surjection):
    '''A uniform dequantization layer.'''

    def forward(self, x):
        z = x.float() + torch.rand_like(x)
        ldj = torch.zeros(x.shape[0])
        return z, ldj

    def inverse(self, z):
        x = z.floor().long()
        return x
```

## H.3  Augmentation

We implement `Augment`, a generative tensor slicing surjection, which may be used to construct e.g. augmented normalizing flows (Huang et al., 2020; Chen et al., 2020).

```python
class Augment(Surjection):
    '''An augmentation layer.'''

    def __init__(self, encoder, split_size):
        super(Augment, self).__init__()
        self.encoder = encoder
        self.split_size = split_size

    def forward(self, x):
        z2, log_qz2 = self.encoder.sample_with_log_prob(context=x)
        z = torch.cat([x, z2], dim=1)
        ldj = -log_qz2
        return z, ldj

    def inverse(self, z):
        x, z2 = torch.split(z, self.split_size, dim=1)
        return x
```

## H.4  Example: Augmented Normalizing Flows

We showcase here the simplicity of implementing an augmented normalizing flow using the SurVAE flow framework. In Listing 1, a simple normalizing flow consisting of 2 coupling layers is constructed. In Listing 2, this is extended by adding an `Augment` surjection, resulting in an augmented flow.

| Listing 1: A basic flow. | Listing 2: An augmented flow. |
|---|---|
| ```Flow(base_dist=Normal((2,)),
    transforms=[

        CouplingBijection(),
        Reverse(),
        CouplingBijection(),
    ])``` | ```Flow(base_dist=Normal((4,)),
    transforms=[
        Augment(Normal((2,)), (2,2)),
        CouplingBijection(),
        Reverse(),
        CouplingBijection(),
    ])``` |

Using the models in Listing 1 and Listing 2, we compare a standard coupling flow with a simple extension using an additional `Augment` layer. We use 4 coupling layers instead of 2 and train models both using identical setups 10000 iterations each. Augmented flows have improved capabilites of modelling data with disconnected components. In Fig. 8, we observe that the augmented flows tend to place their mass more out in a more "clean" fashion and thus demonstrate improved ability to model complicated 2D densities.

Figure 8: Augmented flows show improved capabilities over flows at modelling 2D densities, especially where there are disconnected components. With SurVAE flows, augmented flows are implemented by adding a single surjective augmentation layer to the flow as shown in Listing 2.

# I Experimental Details

We here give more details on the experiments. For further details, see our open-source code[4].

## I.1 Synthetic Data

**Data.** We used 4 synthetic datasets, `checkerboard`, `corners`, `gaussians` and `circles`. For each syntheric dataset, 128000 samples were used as a training set and 128000 more samples as a test set. The `checkerboard` dataset is anti-symmetric, while the 3 others are symmetric.

**Training.** We used the Adam optimizer (Kingma and Ba, 2015) with a learning rate of $10^{-3}$. All models were trained for 10000 iterations (10 epochs) using a batch size of 128.

**Baseline.** The baseline flow is a composition of 4 affine coupling bijections with the ordering reversed in-between. The coupling layers are parameterized by MLPs with hidden units (200,100) and ReLU activations. The base distribution is a standard Gaussian.

**Symmetric AbsFlow.** For the symmetric datasets, AbsFlow uses all the same layers as the baseline. In addition, an `abs` surjection is added, followed by and inverse softplus (`gaussians`) or logit (`checkerboard` and `corners`). In the generative direction, the `abs` surjection randomly samples the sign with equal probabilities. The extra layers contain no parameters, and the AbsFlow thus have the exact same number of parameters as the baseline.

**Anti-Symmetric AbsFlow.** For the anti-symmetric dataset, AbsFlow uses only a single `abs` surjection and a uniform base distribution. In the generative direction, a classifier network learns the probabilities of sampling the sign conditioned on $z$. This classifier network is, like the coupling layer networks, an MLP with (200,100) hidden units and ReLU activations. In this case, the AbsFlow thus has ~$1/4$ the number of parameters.

## I.2 Point Cloud Data

**Data.** We used the `SpatialMNIST` dataset (Edwards and Storkey, 2017). This dataset was constructed by, for each digit in the `MNIST` dataset, sampling 50 points according to the normalized pixel intensities. We used the official code[5] to construct the dataset. We split the dataset into parts of 50000-10000-10000 for training, validation and test (without shuffling). Each data example is a set of 50 2D points which we represent as a tensor of shape (2,50).

**Training.** Both models were trained for 500 epochs using a batch size of 128. We used the Adam optimizer (Kingma and Ba, 2015) with an initial learning rate of $10^{-3}$. The learning rate was warmed up linearly for 2000 iterations and the decayed by 0.995 every epoch. All models were trained using a single GPU for about 40 hours.

**Evaluation.** SortFlow allows exact computation of the likelihood, while the PermuteFlow only allows computation of lower bounds. We evaluated PermuteFlow using the IWBO (importance weighted bound) (Burda et al., 2016) using $k = 1000$ importance samples. PermuteFlow obtains an ELBO of -5.32 PPLL and an IWBO of -5.30 PPLL, while SortFlow obtains an exact log-likelihood of -5.53 PPLL.

**Hyperparameters.** We tuned the dropout rate using the validation set. We considered dropout rates of $\{0.0, 0.05, 0.1, 0.2, 0.3\}$ for both models. We found 0.1 to work best for PermuteFlow, while 0.2 worked best for SortFlow.

**PermuteFlow.** We used a flow of an initial stochastic permutation layer followed by 32 steps with ActNorm layers (Kingma and Dhariwal, 2018) in-between. Each step consisted of 1) an affine coupling bijection which transforms a the first half tensor (1,50) conditioned on the other half (1,50), 2) reversing the order along the spatial dimension, 3) an affine coupling bijection which transforms the first half tensor (2,25) conditioned on the other half (2,25), 4) a stochastic permutation along the point dimension. Each of the coupling bijections are parametersized by Transformer networks (Vaswani et al., 2017) *without* positional encoding. The Transformers used 2 blocks, with $d_{\text{model}} = 64$, $d_{\text{ff}} = 256$ and 8 attention heads.

**SortFlow.** This follows the setup of PermuteFlow, with the following changes: 1) The initial stochastic permutation is replaced by a sorting layer. 2) The stochastic permutations in the flow are swapped with fixed permutations (sampled at random once, before training). 3) The Transformers make use of a learned positional encoding, since the sorting layer enforces a canonical ordering of the points.

## I.3  Image Data

**Data.** We used the `CIFAR-10`, `ImageNet` $32 \times 32$ and `ImageNet` $64 \times 64$ datasets. The `CIFAR-10` dataset comes pre-split in 50000 training examples and 10000 test examples. The `ImageNet` datasets also come pre-split in 1,281,149 training examples and 49,999 validation examples. We use these splits and report results for the test set of `CIFAR-10` and the validations sets of `ImageNet` $32 \times 32$ and `ImageNet` $64 \times 64$.

**Training.** We used the `Adamax` optimizer (Kingma and Ba, 2015) with an initial learning rate of $10^{-3}$ and a batch size of 32. The learning rate was linearly warmed up for 5000 iterations. For `CIFAR-10`, the models were first trained for 500 epochs with the learning rate decayed by 0.995 every epoch. Next, the models were "cooled down" for an additional 50 epochs with a smaller learning rate of $2 \cdot 10^{-5}$. For the `ImageNet` datasets, the models were first trained for 25 epochs (`ImageNet` $32 \times 32$) and 20 epochs (`ImageNet` $64 \times 64$) with the learning rate decayed by 0.95 every epoch. Next, the models were "cooled down" for an additional 2 epochs with a smaller learning rate of $5 \cdot 10^{-5}$. The `CIFAR-10` and `ImageNet` $32 \times 32$ models were trained on a single GPU for about 2 weeks, while the `ImageNet` $64 \times 64$ models were trained using 4 GPUs for about 3 weeks. We provide pre-trained model checkpoints in our open-source code. Note that data augmentation was applied during training of the `CIFAR-10` models, including random flipping and rotations. See code for more details.

**Evaluation.** The `CIFAR-10` models were evaluated using the IWBO (importance weighted bound) (Burda et al., 2016) using $k = 1000$ importance samples. The `ImageNet` models were evaluated using the ELBO (which corresponds to the IWBO with $k = 1$ importance sample).

**Baseline.** For `CIFAR-10` and `ImageNet` $32 \times 32$, the flow uses 2 scales with 12 steps/scale. For `ImageNet` $64 \times 64$, the flow uses 3 scales with 8 steps/scale. Each step consists of an affine coupling bijection (Dinh et al., 2017) and an invertible $1 \times 1$ convolution (Kingma and Dhariwal, 2018). All models are trained with variational dequantization (Ho et al., 2019) and an initial squeezing layer (Dinh et al., 2017) to increase the number of channels from 3 to 12. The coupling bijections are parameterized by DenseNets (Huang et al., 2017).

**MaxPoolFlow.** The MaxPoolFlow uses the exact same setup as the baseline, but replaces the tensor slicing surjection with a max pooling surjection. In the generative direction, we used the simplest possible choice: Each input pixel is equally likely to be copied to any of the pixels in its corresponding $2 \times 2$ patch. The remaining 3 elements are sampled such that the copied value remains the largest: They are set equal to this maximum value minus noise from a standard half-normal distribution (i.e. Gaussian distribution with only positive values). We used simple choices containing *no extra parameters* in order to facilitate more fair comparison. Note that the max pooling layer could be potentially be improved by using more sophisticated choices for the distribution for sampling the remaining elements, $p(\boldsymbol{x}_{-k}|\boldsymbol{z})$, and/or by using a classifier, $p(\boldsymbol{k}|\boldsymbol{z})$, to predict the indices $\boldsymbol{k}$.

# J Additional Samples

Samples from SurVAE flows trained on CIFAR-10, ImageNet $32 \times 32$ and ImageNet $64 \times 64$ using either max pooling or tensor slicing for downsampling are shown in Fig. 9, Fig. 10 and Fig. 11, respectively.

(a) Max Pooling.      (b) No Pooling.

Figure 9: Unconditional samples from SurVAE flows trained on CIFAR-10.

(a) Max Pooling.      (b) No Pooling.

Figure 10: Unconditional samples from SurVAE flows trained on ImageNet $32 \times 32$.

(a) Max Pooling.

(b) No Pooling.

Figure 11: Unconditional samples from SurVAE flows trained on `ImageNet` $64 \times 64$.

## Footnotes

[3]The code is available at `https://github.com/didriknielsen/survae_flows`

[4]https://github.com/didriknielsen/survae_flows

[5]https://github.com/conormdurkan/neural-statistician