[Reviews · NeurIPS 2020]

Review 1

Summary and Contributions: The authors first present traditional latent variable models p(x, z) and deterministic normalizing flows x = f(z) under a common flow framework which incorporates stochastic transitions. The authors then note that a surjection f : Z -> X can be thought of as a deterministic mapping in the direction Z -> X, but a stochastic mapping in the direction X -> Z (the reverse applies for a surjection g: X -> Z), thus allowing natural composition of these surjections under the proposed common flow framework. With this, the authors present common flow operations like slicing and dequantization as surjective transformations, as well as proposing and evaluating surjections for data with symmetries about the origin using an abs operation, exchangeable data using sorting and stochastic permutation operations, and image data with a max pooling operation.

Strengths: Drawing connections between existing parts of the literature is always welcome, and the presentation of a unifying perspective on VAE-type models and deterministic flows under the guise of 'stochastic flows' in Section 2 is neat. Moreover, recognising that surjections are bijective & deterministic in one direction and stochastic in the other allows some insight into common flow transformations (slicing and rounding), and at the very least allows these transformations to now be incorporated more naturally in software implementations. The proposal of novel surjections in 3.1 is a nice step further, and demonstrates that the common framework presented in the preceding sections is worthwhile. The proposed surjections are each motivated by some hypothetical use case, and their effectiveness explored for these tasks in the later experiments. Overall, the distillation of these transformations as distinct modules, each of which specifies i) a forward method, ii) an inverse method, and iii) a likelihood-contribution outlines a nice API, which is both conceptually appealing, and lends itself well to practical implementation. Finally, the empirical evaluation of these surjections involves an interesting set of tasks, and serves to demonstrate that the proposed surjections do indeed prove useful in the ways advertised.

Weaknesses: My main issue is the extent to which a significant amount of the literature is attributed as a 'special case' (51) of the proposed framework. I'm a little concerned that augmented flows (which addressed fundamental limitations in the class of mappings learnable by flows), mixture flows (e.g. CIFs which thoroughly examined the topological limitations of flows), and dequantization (which formally addressed the adaptation of continuous models for discrete data) are labeled as merely 'special cases' (200). All of these are important contributions in their own right, and don't necessarily just drop out of the framework offered here. There's also a sense in which latent variable models which attempt Gaussianization of input data (as exemplified by the VAE with latents of same dimensionality) can already be interpreted as diffusion models, and this diffusion perspective is discussed in Tabak and Vanden-Eijnden 2010. I still think the perspective offered here is useful, I'm just a little worried about the extent to which it maybe overclaims. I do have some concerns as to the ultimate usefulness of the novel surjections. For example, the abs symmetry is neat, but the use case for data with symmetries about the origin is maybe a bit niche. In addition, unless I'm mistaken, the stochastic permutation only 'softly' enforces permutation invariance, instead of making the flow formally invariant to permutation i.e. p(Tx) = p(x) by definition, where T is a permutation. I'd also like to see at least a discussion of the choice of stochastic inverse parameterization in the main text, rather than being relegated to the appendix, and preferably some evaluation of what impact this choice has on the performance\stability of the transformations.

Correctness: The derivations seem correct overall (with some points in additional feedback below). The further details in the appendix are appreciated. The experiments are nicely constructed to demonstrate the potential usefulness of the introduced surjections.

Clarity: The paper is generally well written, with some points which might need to be addressed outlined in the additional feedback below.

Relation to Prior Work: Although recent, 'Stochastic Normalizing Flows' (Wu et al. 2020) proposes flows with stochastic transitions based on a number of well-known MCMC transition kernels, and additionally identifies the variational lower bound as a generalized change of variables formula, as presented in lines 88-106 of this submission. I feel it's important for this work to be cited here, although again I sympathise that the work is very recent. Additionally, 'Deep Unsupervised Learning Using Non-equilibrium Thermodynamics' (Sohl-Dickstein et al. 2015), which trains diffusion models (can be interpreted as stochastic flows) for density estimation using a variational lower bound, also warrants a mention, as well as probably a more thorough discussion of classic diffusion models.

Reproducibility: Yes

Additional Feedback: 33: 'the problem of disconnected components' is a bit vague - what exactly is the problem of disconnected components? I imagine you mean fitting distributions which are supported on disconnected subsets of Euclidean space, but this should probably be explicit. 37: This question is sudden, and could do with a lead-in -- maybe prepend 'these shortcomings motivate the question:'. 58-61: This paragraph reads as if f can be both stochastic and bijective, which is not the case -- perhaps remove 'furthermore'. The definition of a stochastic mapping is also slightly odd: it implicitly assumes the existence of a joint distribution p(x, z), which is not explicitly defined. What I presume is meant by a stochastic mapping is that x = f(z) is equivalent to drawing x ~ p(x | z) defined by some joint p(x, z). Lastly, the definition of a bijection is confusing. In particular, if [x1 = f(z) and x2 = f(z) -> x1 = x2] is not true, then f is not even a valid function. It might be better to take the definition of a surjection in 117-118, and instead define surjections and injections in 58-61, since you can then define bijections using these, and surjections need to be defined for the rest of the paper anyway. I think it's important that this paragraph is clear because stochastic mappings, bijections, and surjections form the backbone of the paper. Eq. (2) is incorrect - the KL term in the ELBO should have a minus sign. I imagine this is just a typo since it's correct in 89 (not to mention the problems we'd have if (2) was true in general). 85 and footnote (1): Why not just use J in the first place? 88-89: 'a single Monte Carlo' -> 'a single Monte Carlo sample' 110-111: Bijective transformations are not required to preserve dimensionality e.g. f(x) = (x, x) is a bijection from R onto its image {(x, y) | x = y} in R^2, with inverse f^{-1}(x, y) = x by definition a bijection from this image back to R. However, f is not a bijection into R^2 which is what I presume is meant here. 113-114: 'alter dimensions' is ambiguous (and sounds a little odd): altering dimensions could refer to transforming existing dimensions of the data, not changing the dimensionality using e.g. a stochastic mapping. 147: Unless I'm mistaken, this describes a flooring rather than a rounding operation? 176-177: stochastic 'inverse' is referred to as 'forward' transformation in 170-171 and Table 2. Table 4: I don't understand the bolding - in each case, Flow++ reports equal or better results, so what does the bold font signify? ---------------------------------------- POST-REBUTTAL UPDATE ---------------------------------------- I'd like to thank the authors for their response. Overall I think the paper puts forward an interesting theoretical perspective and practical framework for future work, and the novel flow layers are a sufficient proof of concept. I've raised my score 6 -> 7, and would like to see the paper accepted.


Review 2

Summary and Contributions: The authors propose a framework of composable bijective and surjective transformations that contains both flows and VAEs. Depending on the direction of the surjection, they allow for exact or approximate estimation of the likelihood. The proposed framework unifies several recent proposals in the normalizing flow literature, and the authors provide several interesting examples of surjective operations such as max value or sorting expressed as SurVAE Flows.

Strengths: To my knowledge, the proposed SurVAE Flows framework is novel and sound/theoretically grounded. It is also presents a significant contribution to the field -- it bridges the gap between flows and VAEs. The insight that surjections can be viewed as deterministic in one direction and stochastic in the other and can therefore be used for this is very well presented and executed (a large set of surjections is considered and expanded upon in the appendix). It unifies several previous approaches into a common framework that will surely spawn further research and will have longer term impact in the field. In addition, the paper is very well presented, which will further enhance this works impact

Weaknesses: Perhaps the biggest weakness might be the limited space in the NeurIPS format -- many details are omitted and/or moved to an appendix. The authors use the limited space well, though. Especially for the larger scale image benchmarks, the proposed framework doesn't lead to marked improvements over baselines. I would be curious about a discussion of this. Computational complexity and runtime are not discussed. I don't think they are larger than for other flows, but it might make sense to mention runtimes/complexities somewhere briefly (for completeness).

Correctness: Yes, to the best of my knowledge

Clarity: The paper is very well written and and very well presented -- a pleasure to read. It is clear and concise and the figures and illustrations support the writing. Well chosen examples illustrate the strength of the method.

Relation to Prior Work: As far as I am aware of it, previous work is discussed and compared/contrasted very well. The related work section is accompanied by an extended related work section in the Appendix and Table 3 in the main paper is an exceptional example for how to put this work into context.

Reproducibility: Yes

Additional Feedback: I would like to congratulate the authors on a very good paper. Some questions: 1) In the derivations, you take limits in function space that lead to expressions of the form `\delta(...) / \delta(...)` -- are these limits well defined and controlled? To make these arguments more rigorous, some function theory/measure theory may be required, though I am happy with the presentation as is. I am more curious whether you thought about this and how these derivations could be made more rigorous. Maybe a sentence/comment in the paper might be in order about this. 2) Regarding point cloud experiments on SpatialMNIST: Your model sounds far more complex than the network considered in the neural statistician, yet your improvements are very small (-5.30 vs -5.37). Could you elaborate on why Neural Statistician holds up so well in your opinion? How do the runtimes compare? Some minor comments: * Thank you for using (author year) citation style -- makes it much more readable than the normal NeurIPS numbering format. * l 58: X and Z cannot be random variables and subsets at the same time. You probably need different names for the random variables. * Tabl 1: give a name to V and E (or mention in the caption) to make the table self contained * l129: B(x) is also referred to as the "preimage" in maths * l157: Eq 3 is referenced but doesn't exist * l157: abbreviations Eq and Table are inconsistent. * some references contain links while others don't ### after rebuttal ### I have read the other reviews and thank the authors for their rebuttal. While some experimental results only show small improvements over baselines, the conceptual/theory contribution and its clear presentation are substantial enough to warrant a strong accept and I maintain my score. In addition to addressing the promised changes (extended comparison to Neural Statistician, clarifications on computational complexity/mathematical details re cancellation of delta functions), I ask the authors to address the related work mentioned by the other reviewers (most notably "Stochastic Normalising Flows" and parts of the derivation in "NICE").


Review 3

Summary and Contributions: The paper presents SurVAE: a framework that unifies normalising flows and VAE’s under a more general framework of estimating likelihoods for surjective transformations. They show that they can explain likelihood evaluation for several methods like flows, vaes, dequantisation, augmented flows and even discrete operations like permutations under this framework, and provide a more modular code design for doing so. They suggest novel SurVAE layers like absolute, max, sort and permutation surjections, and provide preliminary experiments to show that they work in practice. == Post rebuttal update == The authors address including NICE derivation comparison, and image experiments. My score didn't deduct points for this, so I keep the score of 8.

Strengths: - The unification of a lot of ideas in flows and vae under a single framework for likelihood bounds is a significant novel contribution. The theory is simple yet new and well presented. Really liked the derivation in appendix A, shows a single dirac function composition is enough. - The modular framework for composing new flows/vaes using surjective transformations provides a new lens to think of future flow/vae models. They suggest a few new layers themselves like abs, max, sort, permute surjections. Empirically, these perform better when the inductive bias matches, like in the synthetic data experiment and point cloud experiment.

Weaknesses: - The NICE: Non-linear Independent Components Estimation paper has a derivation on page 12 that shows that typical gaussian VAE’s are the same as an augmented flow, and shows how to derive the VAE likelihood bound from the flow likelihood in that scenario. It would be good to include a discussion of this and its comparison to your framework/derivation since it also provides a unifying view of vaes and flows. - The image data experiments (lines 279-294) is slightly weak as the difference with baseline is not much in terms of likelihood and sample quality, so its hard to suggest if max pooling is a better SurVAE block than slicing. Maybe it can be explored from a different lens: since altering latent dimension is one of the motivations of the paper, what it the qualitative difference in the latent space learnt by the two methods? Does having a smaller latent dimension give some other benefits?

Correctness: Yes, claims are correct and empirical methodology is sound

Clarity: Yes, really well written. It uses easy notation, builds up the theory gradually, gives easy to understand examples, which helps a lot as its introducing framework to study the flow/vae models.

Relation to Prior Work: Yes, connection to previous work section is thorough. Only thing I’d suggest referring to is the discussion in the NICE paper page 12 (mentioned above)

Reproducibility: Yes

Additional Feedback:


Review 4

Summary and Contributions: This paper introduces a new kind of probabilistic model somewhere between Variational Autoencoders (VAEs) and normalisaing flows (NF). The aim being to be able to richly model distributions the way normalising flows allows you to do without being always forced to use bijections to do so. This is achieved by having surjections where the transformation is deterministic in one direction stochastic in another. This gives you a tractable likelihood you can optimise with and still have exact likelihoods. The paper also then introduces some novel surjective transformations and show that many existing flows can be expressed in the SurVAE flows framework.

Strengths: This paper introduces a straightforward extension to normalising flows that can allow them to model distributions with interesting symmetries. The method seems fairly novel and I expect many researchers would build upon the work. The evaluation while not exhaustive are very illustrative of the potential of the method.

Weaknesses: The theoretical rationale for why the method works isn't really explored. The modifications to the likelihood, I don't really know what this objective we are optimising is now. Is it a lower-bound? Is it an approximation? How good of an approximation of the likelihood do we learn now? None of this is explored very deeply. The paper would be greatly strengthened if some more grounding was provided for why we should expect the method to work.

Correctness: I don't see any errors or claims that are unjustified in this work. The equations appear to be correct.

Clarity: The paper is very clearly written with well-motivated examples introduced at appropriate places.

Relation to Prior Work: I felt related work was well-covered by not necessarily always explicitly compared against in the experiments.

Reproducibility: Yes

Additional Feedback:

[Author Response · NeurIPS 2020]

We thank all reviewers for their positive and constructive feedback. Reviewers find strengths of the work to be a unified
framework that includes not only flows and VAEs, but also surjections (R1,R2,R3,R4). Some reviewers note that this
framework outlines a modular API that is both appealing and practical (R1,R3) and some note that they expect the
community to build upon the work (R2,R4). We address individual concerns below.

**Reviewer 1:**

- *Special case:* We apologize if our statement came across as dismissive of the previous work. That was certainly
not our intent and we acknowledge that all these works have significant contributions in addressing the short-
comings of flow models. We wanted to suggest that models like Augmented Normalizing Flows, Continuously
Indexed Flows and others fit nicely in our SurVAE Flow framework and can be derived architecturally using
SurVAE blocks. We will rephrase our exposition to better reflect this.

- *Diffusion perspective:* Thanks for bringing this to our attention. We will add a reference to (Tabak and
Vanden-Eijnden 2010) in the updated version.

- *Niche problems:* Our intention was to demonstrate that common (surjective) operations such as abs, max, sort,
etc. can be useful transformations in generative models. This showcases the fact that the framework can be
used to model data that would be more difficult to model with regular flows and therefore suggests that further
development of novel SurVAE layers might be worthwhile.

- *Permutation invariance:* The resulting model is permutation invariant $p(Tx) = p(x)$. Computing the true
likelihood requires an intractable sum over all permutations. When the likelihood is approximated using
Monte Carlo samples, you will get an unbiased approximation of the true likelihood. However, this unbiased
approximation is also permutation invariant (when the random seed is not fixed).

- *Stochastic inverse parameterization:* We will (as much as the page limit allows) add a discussion on this in the
main text. In our experiments we used only very simple stochastic inverse distributions (as discussed in App.
J: Experimental Details).

- *Additional references:* We will add references to (Wu et al. 2020) and (Sohl-Dickstein et al. 2015). Thanks for
pointing these out.

**Reviewer 2:**

- *Image experiments:* In the image experiments, the max pooling surjection performs similarly to the slicing
surjection (which corresponds to the regular multi-scale architecture). We do not claim that max pooling
surjection gives improvements over baselines. Rather, we demonstrate that max pooling is a viable alternative
for constructing multi-scale flows. Note that we use a max pooling layer with a simple stochastic inverse with
no extra parameters (see App. J). Improvements can likely be made by using more sophisticated choices.

- *Computational complexity:* The reviewer is correct, the computational complexity is similar to other flow
models (and always the same for models that we explicitly compare). We will add more details on this.

- *Delta functions:* We are unsure exactly which derivations are being referred to, but if it is the derivations of the
surjective layers in App. E, F, G, H, more thorough derivations, leading to the same final results, can be made
using the following steps: 1) Specify Gaussian distributions with standard deviation $\sigma$ in place of the Dirac
deltas. 2) Write out the likelihood contribution term and cancel terms in the numerator and denominator to get
rid of $\sigma$. 3) Take the limit as $\sigma \to 0$, to obtain the likelihood contribution for the surjective transformation.

- *Neural Statistician:* The reviewer raises a nice point and it will be interesting to study this further. However,
providing an informed discussion on this question requires a careful and controlled set of experiments with
clearly defined tasks which we shall try to accommodate in our revision.

- *Experiment bolding:* In the table, we are comparing the baseline to the same model but with a max pooling
surjection. The bold font thus only compares the performance of these two models. We apologize for this
confusion and will clarify this in the revised version.

**Reviewer 3:**

- *NICE:* Thanks for pointing out the derivation in NICE. We will include discussion of this.

- *Image experiments:* See Point 1 for Reviewer 2. Reviewer 3 makes a valid point about qualitatively comparing
the latent spaces learnt using max pooling vs. slicing. This is worth exploring as a future direction.

**Reviewer 4:** It seems you forgot to include the Weaknesses section, which makes it hard for us to address your concerns.
However, we hope that the other reviews, together with our response, will alleviate some of your concerns.

**All:** We have also noted all minor comments and will use these to improve the paper. Thanks again for your work.

[Meta-Review · NeurIPS 2020]

The authors introduce a novel conceptual framework that unifies normalizing flows and VAEs and includes many other existing models and modules such as augmented flows and variational dequantization. The framework involves thinking about generative models in terms of the type of mapping they use to go from the observation to the latents and vice versa. This turns out to be fruitful because it immediately makes apparent the gap between flows, which use deterministic mappings in both directions, and VAEs, which use stochastic mappings. The authors fill this gap by introducing surjective models/components which are deterministic in one of the directions and stochastic in the other, and proceed to derive several instances of these, e.g. max and sort surjections. The reviewers found the paper insightful and praised the quality of exposition. They thought the experiments were sufficient to demonstrate that the proposed components work, even if they did not always lead to an improvement over the existing architectures. By describing probabilistic modules in terms of the forward and backward mappings as well as their contributions to the log-likelihood (or the ELBO), the paper provides a clear template for implementations that enable easy composability. This could be an influential contribution. Similarly, the idea of surjective components along with the provided examples might spur derivation of more such modules, expanding the range of easy-to-use primitives for probabilistic modeling.